# Early redox activities modulate *Xenopus* tail regeneration

Fernando Ferreira [1,2], VijayKrishna Raghunathan [3,4,5], Guillaume Luxardi[1], Kan Zhu [1] & Min Zhao [1,6]

Redox state sustained by reactive oxygen species (ROS) is crucial for regeneration; however, the interplay between oxygen ($O_2$), ROS and hypoxia-inducible factors (HIF) remains elusive. Here we observe, using an optic-based probe (optrode), an elevated and steady $O_2$ influx immediately upon amputation. The spatiotemporal $O_2$ influx profile correlates with the regeneration of *Xenopus laevis* tadpole tails. Inhibition of ROS production but not ROS scavenging decreases $O_2$ influx. Inhibition of HIF-1α impairs regeneration and stabilization of HIF-1α induces regeneration in the refractory period. In the regeneration bud, hypoxia correlates with $O_2$ influx, ROS production, and HIF-1α stabilization that modulate regeneration. Further analyses reveal that heat shock protein 90 is a putative downstream target of HIF-1α while electric current reversal is a de facto downstream target of HIF-1α. Collectively, the results show a mechanism for regeneration via the orchestration of $O_2$ influx, ROS production, and HIF-1α stabilization.

[1] Department of Dermatology, Institute for Regenerative Cures, University of California, Davis 95616 CA, USA. [2] Departamento de Biologia, Centro de Biologia Molecular e Ambiental (CBMA), Universidade do Minho, Braga 4704, Portugal. [3] Department of Basic Sciences, College of Optometry, University of Houston, Houston 77204 TX, USA. [4] The Ocular Surface Institute, College of Optometry, University of Houston, Houston 77204 TX, USA. [5] Department of Biomedical Engineering, Cullen College of Engineering, University of Houston, Houston 77204 TX, USA. [6] Department of Ophthalmology, Institute for Regenerative Cures, University of California, Davis 95817 CA, USA. Correspondence and requests for materials should be addressed to F.F. (email: id3955@alunos.uminho.pt) or to M.Z. (email: minzhao@ucdavis.edu)

Large-scale injuries undergo repair via regeneration in highly regenerative taxa, such as amphibians. At a molecular level, biochemical signaling pathways, such as Wnt, FGF, BMP, and TGF are necessary for appendage regeneration[1,2]. Reactive oxygen species (ROS) have recently been identified as pivotal signaling cues for regeneration in a plethora of regeneration models. ROS attract immune cells to the site of injury, regulate Wnt, FGF, and MAP kinase pathways, influence proliferation and differentiation, and modulate bioelectric activities[3–8].

That oxygen ($O_2$) potently affects cell signaling and behavior is now well established[9,10]. Hypoxia (partial pressure of oxygen ($pO_2$) of less than 2% (14 mmHg or 19 hPa)) is required for normal development[9], maintenance of pluripotency in stem cells niches[11], angiogenesis[12], and is a common feature of malignant tumors[13]. Hypoxia exerts most of its effects through the ubiquitous hypoxia-inducible factor (HIF)-1α, a master regulator of $O_2$ homeostasis. In normoxia, prolyl residues in the $O_2$-dependent degradation domain of HIF-1α are hydroxylated by prolyl hydroxylases (PHD), tagging them for degradation. Hypoxia prevents this, allowing HIF-1α to dimerize with HIF-1β in the nucleus and together bind to hypoxia-responsive elements (HRE) of dozens of target genes[10].

While extensive evidence demonstrates that HIF-1α is upregulated in and required for wound healing in vitro and in vivo[14–17], its role in regeneration remains poorly understood. A recent study found that HIF-1α is necessary for and sufficient to induce mouse ear hole regeneration, perhaps through the regulation of stem cell behavior[18]; although, the exact mechanisms are less understood. In addition to stem cell niches, HIF-1α is commonly present in other mesenchymal condensations, such as limb bud, somites, and cancer[13,19,20]. Interestingly, the regeneration bud is a mesenchymal-like structure with stemness and proliferation capabilities[2].

Altogether, discrete redox players, mainly ROS, are crucial for regeneration; however, an integrative interplay between $O_2$, ROS, and HIF-1α during regeneration remains utterly elusive. Here, using the *Xenopus laevis* tadpole tail regeneration model[2,21], we hypothesized that an injury-induced $O_2$ influx fuels local ROS production, setting a permissive hypoxia in the regeneration bud to stabilize HIF-1α and subsequently modulate regeneration. First, we observe a close correlation between $O_2$ influx and regeneration. Next, we demonstrate that the sustained influx of $O_2$ fuels ROS production that is required for regeneration. Further, we demonstrate that the hypoxia-stabilized and ROS-independent HIF-1α is necessary for and sufficient to induce regeneration. Finally, we identify heat shock protein (HSP) 90 and electric current ($J_I$) reversal as downstream targets of HIF-1α. Together, our data show that the integrative interplay of these disparate redox players is critical to regeneration.

## Results

### Injury-induced extracellular $O_2$ influx dynamically correlates with regeneration.
Atmospheric $pO_2$ is higher than corporeal (arterial, venous and tissue) $pO_2$[9], which would theoretically lead to an $O_2$ influx down its chemical gradient upon barrier breaking by injury or disease. To provide evidence for this putative influx and to select key spatiotemporal points to test our hypothesis, we started by mapping the extracellular $O_2$ flux during regeneration. $O_2$ fluxes during the three phases of regeneration—wound healing, regeneration bud formation, and regenerative outgrowth[2] (Fig. 1a)—were measured using an optic-based probe (optrode) that quantifies $O_2$ via fluorescence quenching (Supplementary Fig. 1a–d). Before amputation, baseline $O_2$ flux was $-4.26 \pm 0.78$ pmol cm$^{-2}$ s$^{-1}$ in the tail tip (mean ± s.e.m.; $n = 10$ biological replicates; Fig. 1b). The direction (influx) and magnitude

(relatively low compared with uptake in gills) of the flux are in accordance with cutaneous respiration in amphibians[22] (Supplementary Fig. 2). Upon amputation, barrier breaking led de facto to a significant increase in $O_2$ influx of >150% (to $-11.03 \pm 1.85$ pmol cm$^{-2}$ s$^{-1}$ at 5 min post-amputation (minpa), $n = 8$, $p = 0.008$; Fig. 1b). The $O_2$ gradient established upon injury generated an $O_2$ sink at the amputation plane. Thus, we descriptively termed this part of the temporal profile curve as 'slope' (indicated as 'S'; Fig. 1b). The temporal profile presented then a plateau, followed by a positive curve, both correlating with the progression of regeneration. Interestingly, $O_2$ influx did not significantly decrease following wound epithelium formation by 6 h post-amputation (hpa) (from $-11.03 \pm 1.85$ pmol cm$^{-2}$ s$^{-1}$, $n = 8$, to $-8.60 \pm 1.20$ pmol cm$^{-2}$ s$^{-1}$, $n = 13$, $p = 0.262$; Fig. 1a–c). Instead, from 5 minpa to 48 hpa, $O_2$ influx stabilized ($p \gg 0.05$ for all comparisons), correlating with the regeneration bud formation phase (Fig. 1). We thereby termed this part of the curve as 'plateau' (indicated as 'P'; Fig. 1b). At 24 hpa there was a peak in $O_2$ influx (Fig. 1b, c) that was significantly higher than 6 hpa. Peak $O_2$ influx is probably due to the beginning of cell proliferation and bud maturation at 24 hpa[23], both of which increase the demand for $O_2$. Beyond 48 hpa, a final shift resolved the significant $O_2$ influx by 72 hpa (to $-5.04 \pm 0.77$ pmol cm$^{-2}$ s$^{-1}$, $n = 8$, $p = 0.490$; Fig. 1b). When regeneration was complete (7 days), $O_2$ influx returned to a magnitude even lower than what was observed at baseline ($-0.60 \pm 0.26$ pmol cm$^{-2}$ s$^{-1}$, $n = 8$, $p = 0.001$; Fig. 1b); this may be attributed to greater surface area-to-volume ratio with animal growth. We therefore termed this part of the curve as 'baseline' (indicated as 'B'; Fig. 1b) and it correlated with the regenerative outgrowth phase (Fig. 1).

The spatial profile showed no consistent pattern overall, except for often higher $O_2$ influx in the spinal cord in comparison with the dorsal fin (Supplementary Fig. 1e). This spatial-dependency occurred between 3 and 24 hpa, when a semi-circular bud was prominent. This might point to a geometric effect, i.e., an artefactual amplification of the spinal cord $O_2$ influx reading resultant from the inclusion of contiguous bud and dorsal fin influxes (Supplementary Fig. 1a, b).

Together the data demonstrate that the injury created an $O_2$ sink that increases $O_2$ influx and the extracellular flux profile dynamically correlated with the progression of regeneration, suggesting the bud at 6 and 24 hpa (within plateau part) as key spatiotemporal points to test the proposed hypothesis.

### Extracellular $O_2$ influx correlates with regeneration efficiency.
*X. laevis* has an intriguing age-dependent refractory period[24] that, together with the regenerative period, permits to readily investigate limiting and stimulating factors without changing the regeneration model. With this advantage, we examined whether the magnitude and/or direction of $O_2$ flux differ within the refractory period at 6 and 24 hpa. First, we verified that amputation in the refractory period (stage (st.) 45–46) impairs regeneration. The frequency of full phenotypes (tadpoles with complete tail regeneration) decreased >40-fold in comparison to amputation in regenerative (st. 40–41) tadpoles (from 59–1%) and the frequency of none phenotypes (tadpoles without tail regeneration) increased from 0 to 48%. Overall, consistent with previous reports[8,23,24], regeneration was significantly impaired with a reduction in the regeneration index (RI, 0–300; computed from frequencies of phenotypes; Eq. 1) from 257 ($n = 47$) to 84 ($n = 59$, $p < 0.0001$; Fig. 2a, b).

$O_2$ flux direction remained unchanged, with influx observed in all spatiotemporal points (Fig. 2c, Supplementary Fig. 3a). Remarkably, the magnitude of $O_2$ influx in refractory period tails more than doubled compared with regenerative tails. At 6 hpa, $O_2$

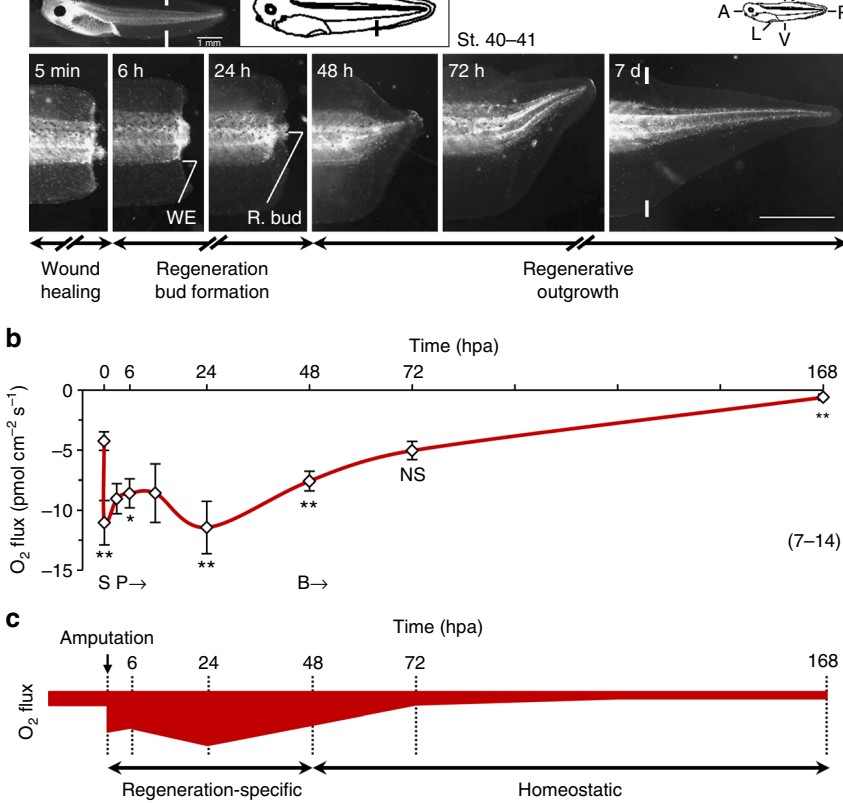

**Fig. 1** Extracellular $O_2$ flux dynamically correlates with regeneration. **a** Regeneration time-lapse and phases of a representative tadpole tail amputated at st. 40–41. Major regeneration structures annotated: wound epithelium (WE) and regeneration bud (R. bud). Photomicrographs are displayed in the same orientation as the whole-organism anteroposterior (A/P), dorsoventral (D/V), and left–right (L/R) axes (top left scheme; applies to subsequent figures). White solid lines: amputation plane (black solid lines in schematic tadpole); scale bars: 1 mm. **b** Temporal profile of $O_2$ flux in the regeneration bud in MMR 0.1× (control). Negative values are net influx (applies to subsequent figures). Profile is descriptively divided into three parts: S slope; P plateau, and B baseline. **c** Diagrammatic representation of the temporal dynamic of $O_2$ influx during regeneration. Profile is divided into two phases: regeneration-specific and homeostatic. Magnitudes are not at absolute scale but present the relative dynamics in temporal directions and magnitudes. Statistical analyses were performed by unpaired Student's *t*-test (two-tailed *p*-value). The data are presented as mean ± s.e.m. *n* biological replicates indicated in brackets. NS non-significant; \**p* < 0.05; \*\**p* < 0.01

influx in the bud significantly increased 124% (from $-8.60 \pm 1.20$ pmol cm$^{-2}$ s$^{-1}$, $n = 13$, to $-19.27 \pm 1.75$ pmol cm$^{-2}$ s$^{-1}$, $n = 14$, $p < 0.0001$; Fig. 2c) and at 24 hpa, similarly, influx significantly increased 140% (from $-11.43 \pm 2.18$ pmol cm$^{-2}$ s$^{-1}$, $n = 14$, to $-27.49 \pm 2.96$ pmol cm$^{-2}$ s$^{-1}$, $n = 13$, $p = 0.0002$; Fig. 2c). The spatiotemporal profile showed a similar increase in $O_2$ influx in the spinal cord (e.g., 123% increase at 6 hpa), while a smaller increase was observed in the dorsal fin (non-significant at 6 hpa) (Supplementary Fig. 3a).

Regenerative (st. 40–41) tadpoles are smaller than refractory period (st. 45–46) ones[25]. Thus, a difference in the magnitude of $O_2$ influx could be explained, in part, by the higher surface area-to-volume ratio (Supplementary Fig. 3). This ratio and the $O_2$ uptake are, theoretically, inversely proportional. The absent (at 6 hpa) and lower (at 24 hpa) disparities of $O_2$ influx in dorsal fins of regenerative and refractory periods (Supplementary Fig. 3a) point, however, to a more bud-specific and therefore regeneration-specific $O_2$ influx. To discard the size effect of the buds we measured their area at 6 hpa. Regenerative and refractory buds are similar in size (Supplementary Fig. 3b, c), implying that the $O_2$ influxes are independent of bud size.

Altogether, $O_2$ flux correlates with and predicts regeneration efficiency.

**$O_2$ influx is regeneration-specific and a conserved response to injury.** With the demonstration of an injury-induced, steady, and long-lasting $O_2$ sink (elevated $O_2$ influx), two questions arise: is the $O_2$ influx profile regeneration-specific? and, is the elevated $O_2$ influx a conserved/universal response to injury? For the first question, correlation of $O_2$ influx to regeneration efficiency is a robust indication of its causality. To further address the question, we mapped $O_2$ flux during fin wound healing. As in regeneration, there was a significant injury-induced $O_2$ sink (109% increased $O_2$ influx magnitude upon wounding, $p = 0.022$; Supplementary Fig. 4a, d, e) that largely surpassed measured cutaneous respiration needs (Supplementary Fig. 2). The wound's temporal profile correlated with fin healing (Supplementary Fig. 4b, c, e), and was similar to the regeneration profile. However, the plateau part was about 3 h in the wound healing profile compared to 48 h in regeneration, suggesting a correlation between plateau duration and required healing and thus to a regeneration-specific response.

For the second question, we measured $O_2$ flux in a single cell wound model (oocytes[26]) and in a higher *taxon*, the mouse skin wound model. Excitingly, $O_2$ influx significantly elevated in both models upon wounding: 29% in oocytes (from $-14.09 \pm 2.45$ to $-18.14 \pm 2.57$, $n = 6$, $p = 0.003$; Supplementary Fig. 5a–c) and 61% in mice skin (from $-32.39 \pm 3.41$ to $-52.15 \pm 8.04$, $n = 4$,

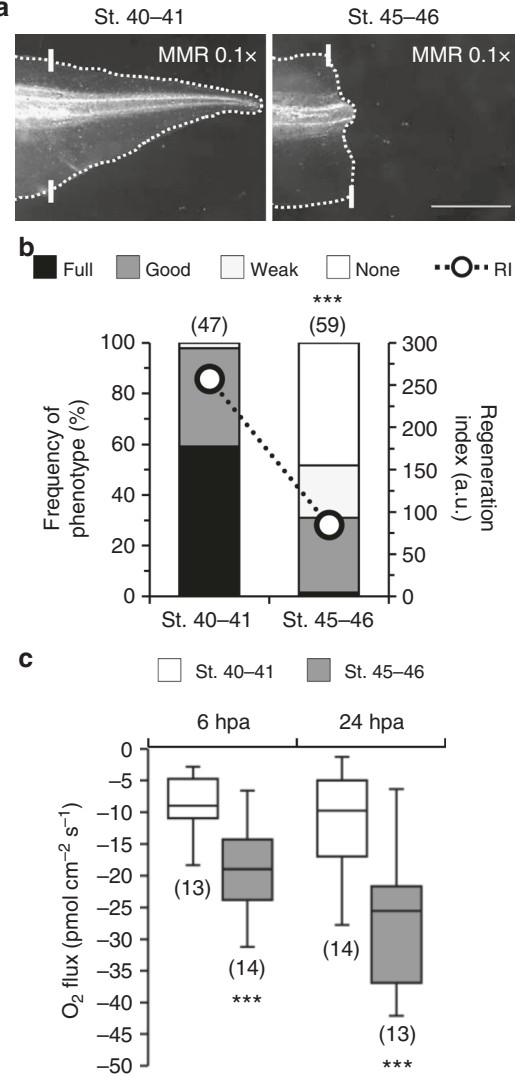

**Fig. 2** Extracellular $O_2$ influx predicts regeneration efficiency.
**a** Representative tails at 7 dpa in MMR 0.1× from tadpoles amputated in regenerative (st. 40–41) or refractory period (st. 45–46). White solid lines: amputation plane; scale bar: 1 mm. **b** Qualitative and quantitative analyses of regeneration efficiency for the different conditions tested. RI regeneration index; a.u. arbitrary units. **c** $O_2$ flux measured in two different time-points in the bud of regenerative or refractory period tadpoles. Statistical analyses were performed by Fisher's exact test **b**, or unpaired Student's $t$-test (both two-tailed $p$-value) **c**. The data are presented as median ± min to max (with outliers). $n$ biological replicates indicated in brackets. ***$p < 0.001$

$p = 0.033$; Supplementary Fig. 5d–f). Collectively, the similarity in responses observed regardless of the wound model and site suggest that elevated $O_2$ influx is a conserved response to injury. This applies to the cases where an $O_2$ gradient and consequent $O_2$ sink can be established upon wounding.

**Exogenous $O_2$ fuels ROS production that is required for regeneration**. To identify mechanisms underlying the correlation between $O_2$ influx and regeneration efficiency, we determined whether ROS production is dependent on the influx of exogenous $O_2$, i.e., of $O_2$ not from the circulatory system (endogenous $O_2$). For this, using diphenyleneiodonium (DPI) we inhibited the NADPH oxidase family[27] and measured the extracellular $O_2$

fluxes in regenerative tadpoles. We and others have demonstrated that DPI decreases ROS production during tail regeneration[6,8]. First, we verified the necessity of ROS in regeneration. DPI-treated tadpoles had a 12-fold reduction in the frequency of full phenotypes (from 73 to 6%); whereas, the frequency of none phenotypes increased from 0 to 31%. Consistent with previous studies[6,8], overall regeneration was significantly abolished (RI reduced from 273, $n = 19$, to 106, $n = 16$, $p < 0.0001$; Fig. 3a, b).

DPI-treated tadpoles showed a significant decrease in $O_2$ influx both at 6 hpa (114% reduction, from $-12.03 \pm 1.24$, $n = 15$, to $-5.63 \pm 1.11$ pmol cm$^{-2}$ s$^{-1}$, $n = 16$, $p = 0.0006$; Fig. 3c) and 24 hpa (172% reduction, from $-17.43 \pm 2.12$ pmol cm$^{-2}$ s$^{-1}$, $n = 14$, to $-6.40 \pm 1.01$ pmol cm$^{-2}$ s$^{-1}$, $n = 17$, $p = 0.0002$; Fig. 3c).

Next, we investigated whether depletion of ROS had a similar effect in $O_2$ influx in the refractory period. Previously, we demonstrated that hydrogen peroxide ($H_2O_2$) is the key ROS necessary for and sufficient to induce regeneration[8], leading us to hypothesize that ROS production is deficient in refractory period tails. To determine if this was indeed true, we performed semi-quantitative imaging of ROS using HyPer transgenic tadpoles that have a constitutive genetically encoded $H_2O_2$-specific sensor[28]. Refractory period tadpoles presented significantly lower $H_2O_2$ levels than regenerative tadpoles at both 6 and 24 hpa (Supplementary Fig. 7a–c), proving the assumption. The use of a general ROS-sensitive dye in wild-type tadpoles also showed a reduction in overall ROS levels in the refractory period, validating HyPer results (Supplementary Fig. 7d, e). We next hypothesized that, due to lower $H_2O_2$ levels, DPI would not robustly affect the magnitude of $O_2$ influx. Indeed, treatment with DPI for 6 hpa did not significantly affect $O_2$ influx in comparison to vehicle-control (from $-20.63 \pm 1.71$, $n = 17$, to $-19.07 \pm 2.22$ pmol cm$^{-2}$ s$^{-1}$, $n = 14$, $p = 0.576$; Fig. 3d) in refractory period tails. However, treatment with DPI for 24 hpa significantly decreased $O_2$ influx (43% reduction, from $-21.34 \pm 1.88$, $n = 14$, to $-14.93 \pm 0.83$ pmol cm$^{-2}$ s$^{-1}$, $n = 15$, $p = 0.006$; Fig. 3d). This reduction in influx was fourfold lesser than the reduction observed with regenerative tadpoles (172%), further highlighting that ROS production in the refractory period was deficient.

To decouple enzymatic ROS production from ROS *per se*, we used the antioxidant trolox and quantified regeneration and measured the $O_2$ flux in the regenerative period. We and others demonstrated that antioxidant decrease ROS production and impair tail regeneration[6,8]. Similarly, trolox impaired regeneration in this study. The frequency of full phenotypes almost halved (1.7-fold decrease, from 69 to 40%) and the frequency of none phenotypes increased from 0 to 17%. Overall, regeneration was significantly impaired (RI reduced from 264, $n = 83$, to 180, $n = 96$, $p < 0.0001$; Fig. 3e, f). Trolox-treated tadpoles showed non-significant shifts in $O_2$ influx at 6 hpa (from $-8.92 \pm 2.09$, $n = 8$, to $-9.50 \pm 1.56$ pmol cm$^{-2}$ s$^{-1}$, $n = 11$, $p = 0.824$; Fig. 3g) or 24 hpa (from $-18.42 \pm 2.10$ pmol cm$^{-2}$ s$^{-1}$, $n = 10$, to $-17.09 \pm 2.27$ pmol cm$^{-2}$ s$^{-1}$, $n = 10$, $p = 0.671$; Fig. 3g).

Altogether, ROS is a probable pathway by which $O_2$ influx correlates with regeneration efficiency in the regenerative tadpoles, but not likely in the refractory period tadpoles due to intrinsically impaired ROS levels.

**HIF-1α is necessary for and sufficient to induce regeneration**. Next, we sought downstream effectors capable of mediating ROS-driven $O_2$ influx and regeneration. Steady and long-term $O_2$ influx and ROS production are likely to influence the $pO_2$ in the bud. Thus, we tested whether the master mediator of hypoxia, HIF-1α, is necessary for regeneration. For this, we used echino-mycin, a small-molecule DNA-binding inhibitor that precludes HIF-1α from binding to HRE, thus inhibiting its action[29].

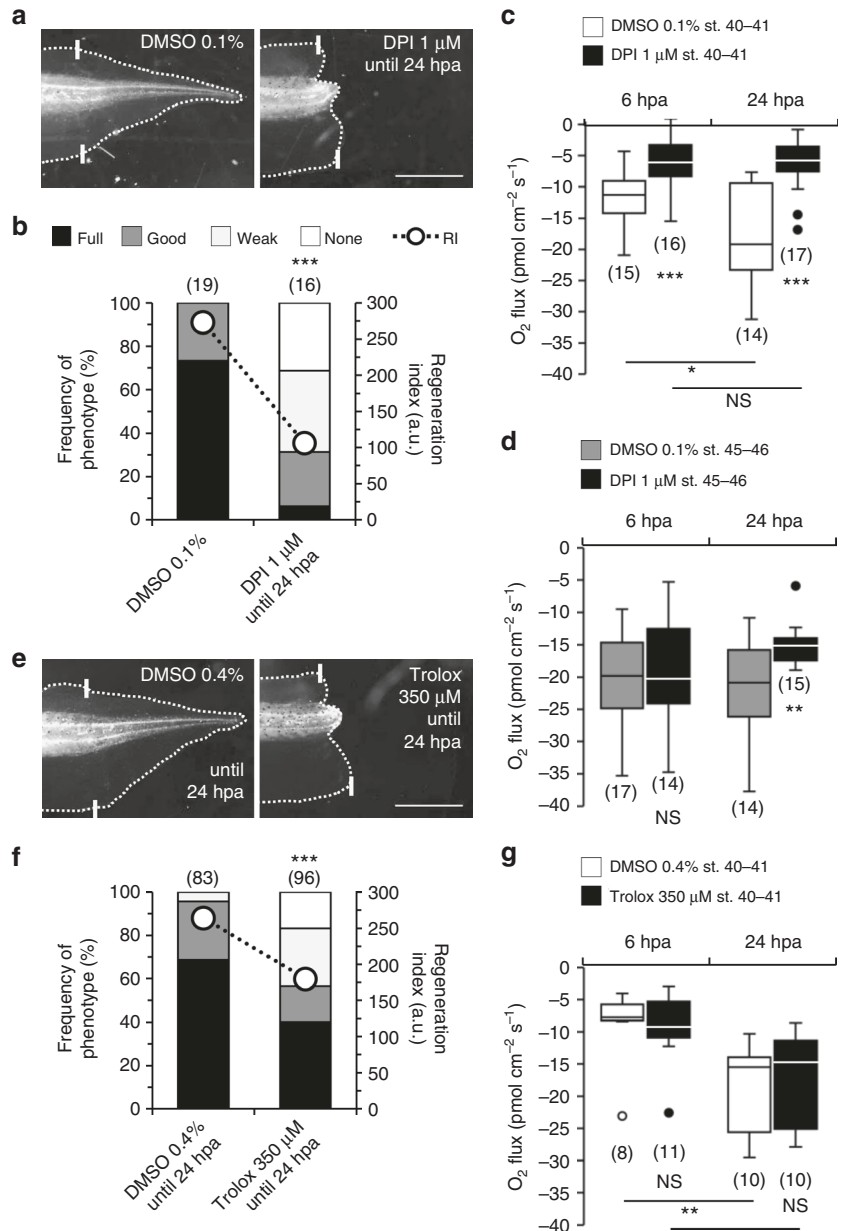

**Fig. 3** Exogenous $O_2$ fuels ROS production that is necessary for regeneration. **a–d** $O_2$ influx fuels ROS production in regenerative but not likely in refractory period tadpoles. **a** Representative tails at 7 dpa in vehicle-control and pharmacological treatment from tadpoles amputated at st. 40–41. **b** Qualitative and quantitative analyses of regeneration efficiency for the different conditions tested. Most tadpoles from $O_2$ flux measurements. RI regeneration index. Stacked bars legend applies to **f**. $O_2$ flux measured in two different time-points in the bud of vehicle-control and pharmacological treatment from tadpoles amputated in regenerative **c** or refractory period **d**. **e–g** The magnitude of $O_2$ influx is independent of ROS per se. **e** Representative tails at 7 dpa in vehicle-control and pharmacological treatment from tadpoles amputated at st. 40–41. **f** Qualitative and quantitative analyses of regeneration efficiency for the different conditions tested. Tadpoles from $O_2$ flux measurements included in quantification. **g** $O_2$ flux measured in two different time-points in the bud of vehicle-control and pharmacological treatment from tadpoles amputated at st. 40–41. White solid lines: amputation plane; scale bar: 1 mm; a.u. arbitrary units. Statistical analyses were performed by Fisher's exact test **b**, **f**, or unpaired Student's $t$-test (both two-tailed $p$-value) **c**, **d**, **g**. Data are presented as median ± min to max (with outliers). $n$ biological replicates indicated in brackets. NS non-significant; *$p < 0.05$; **$p < 0.01$; ***$p < 0.001$

Echinomycin-treated tadpoles had dramatically abrogated regeneration presenting only weak (tadpoles with incomplete or abnormal tail regeneration; 11% frequency) and none (89% frequency) phenotypes. Overall, RI reduced from 248 ($n = 48$) to 11 ($n = 55$, $p < 0.0001$; Fig. 4a, b). The first 24 hpa are more likely to present regeneration-specific events (Fig. 1c). Thus, we next performed an extensive exposure screen to determine the exact time-window of requirement for HIF-1α. First, we exposed the tadpoles to echinomycin for 24 hpa. This treatment was as

penetrant as continuous exposure to HIF-1α inhibitor throughout regeneration (RI reduced to 19, $n = 48$, $p < 0.0001$; Fig. 4b). A refined exposure showed that echinomycin until 15 minpa was still equally penetrant with only 10% weak phenotypes and the remainder 90% presenting a none phenotype (RI reduced from 228, $n = 47$, to 10, $n = 47$, $p < 0.0001$; Fig. 4a, c). The complementary approach showed that echinomycin either from 1 or 2 hpa until the end of regeneration (7 days) impaired regeneration in an exposure-dependent way. Specifically,

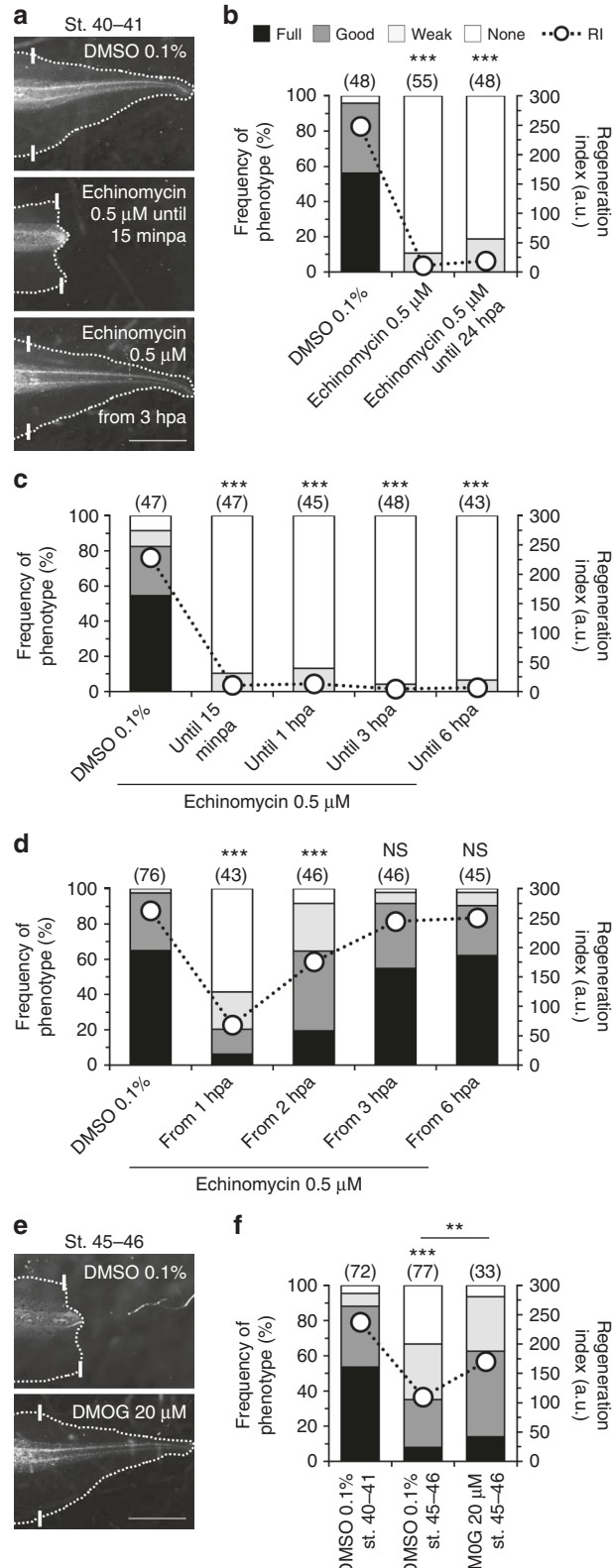

**Fig. 4** HIF-1α is necessary for and sufficient to induce regeneration. **a–d** Loss of regeneration by HIF-1α inhibition and temporal requirement for HIF-1α activity during regeneration. **a** Representative tails at 7 dpa in vehicle-control and pharmacological treatment from tadpoles amputated at st. 40–41. **b–d** Qualitative and quantitative analyses of regeneration efficiency for the different conditions tested. RI regeneration index. Stacked bars legend in **b** applies to **c**, **d**, and **f**. **e**, **f** Induction of regeneration by HIF-1α stabilization. **e** Representative tails at 7 dpa in vehicle-control and pharmacological treatment from tadpoles amputated in refractory period. **f** Qualitative and quantitative analyses of regeneration efficiency for the different conditions tested. White solid lines: amputation plane; scale bar: 1 mm; a.u. arbitrary units. Statistical analyses were performed by Fisher's exact test (two-tailed p-value). n biological replicates indicated in brackets. NS non-significant; **p < 0.01; ***p < 0.001

slightly decreased (from 65 to 55%) and the frequency of good phenotypes (tail regeneration with minor defects) slightly increased (from 33 to 37%) in comparison to vehicle-control (RI from 263, $n = 76$, to 244, $n = 46$, $p = 0.347$; Fig. 4a, d). These results suggest that HIF-1α is essential for regeneration within the first 3 hpa.

Secondary validation of these results was performed using the alternative drug chetomin, a small-molecule transcription inhibitor of HIF-1α[30]. Chetomin also robustly inhibited regeneration and supported the observation that early stabilization of HIF-1α is essential for regeneration (Supplementary Fig. 9a, b). Drug-enhanced stabilization of HIF-1α with the PHD inhibitor dimethyloxallyl glycine (DMOG)[31] showed no effect in regenerative tadpoles (Supplementary Fig. 9a, c).

Next, we attempted to induce regeneration in the refractory period by stabilizing HIF-1α with DMOG. Compared with vehicle-control, DMOG-treated tadpoles almost doubled the frequency of full phenotypes (1.8-fold increase, from 8 to 14%) and decreased the frequency of none phenotypes by fivefold (from 33 to 7%). Overall, regeneration was significantly induced in refractory period by stabilization of HIF-1α (RI from 110, $n = 77$, to 170, $n = 33$, $p = 0.007$; Fig. 4e, f, Supplementary Fig. 9d) demonstrating the importance of HIF-1α stability. Short-term exposure to DMOG (until 24 hpa) did not induce regeneration, despite a slight increase in RI (to 126, $n = 56$, $p = 0.326$; Supplementary Fig. 9e).

Collectively, data demonstrate that HIF-1α is necessary for and sufficient to induce regeneration, suggesting HIF-1α as a candidate pathway downstream of ROS and by which $O_2$ influx correlates with regeneration efficiency in both regenerative and refractory period tadpoles.

**Hypoxia correlates with HIF-1α stabilization that in turn regulates regeneration.** Highest HIF-1α activity was delimited within 1 hpa. Subsequently, we tested whether ROS and refractory period affect hypoxia and HIF-1α stabilization in that time-window. First, using the hypoxia marker pimonidazole[32], we determined the level of hypoxia in the wound epithelium and prospective regeneration bud under all conditions. For a semi-quantitative analysis, cells proximal to the amputation plane were dissociated for flow cytometry. DPI-treated tadpoles had a significant reduction (by 41%) in the overall hypoxia compared with vehicle-control ($n = 3$ flows of 60 specimens, $p = 0.027$; Fig. 5a, b). In the refractory period, hypoxia was dramatically reduced by 70%, compared with the stage control ($n = 3$ flows of 60 specimens, $p = 0.032$; Fig. 5a, b, Supplementary Fig. 13d–f). No significant difference in hypoxia was observed in tadpoles with drug-stabilized HIF-1α (Fig. 5b, Supplementary Fig. 13g).

treatment with echinomycin from 2 hpa decreased the frequency of full phenotypes by 3.4-fold (from 65 to 19%) and increased the frequency of none phenotypes from 0 to 8% (RI reduced from 263, $n = 76$, to 176, $n = 46$, $p < 0.0001$; Fig. 4d). Treatment with echinomycin from 3 hpa or beyond did not significantly impair regeneration. From 3 hpa, the frequency of full phenotypes

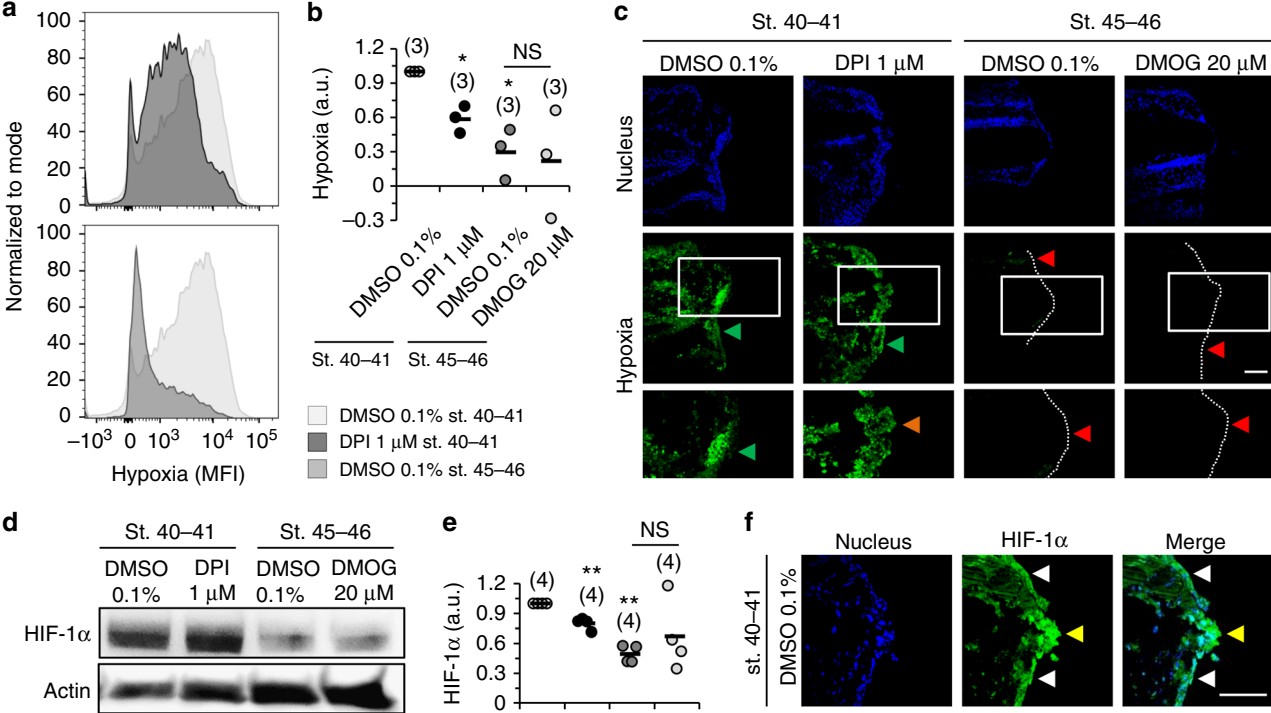

**Fig. 5** Hypoxia and HIF-1α stabilization correlate with regeneration efficiency. **a–c** Hypoxia is significantly affected by depleted ROS and dramatically affected in refractory period. **a** Representative flow cytograms of DMSO 0.1% st. 40–41 vs. DPI 1 μM st. 40–41 (top panel) and DMSO 0.1% st. 40–41 vs. DMSO 0.1% st. 45–46 (bottom panel) at 1 hpa. MFI mean fluorescence intensity. **b** Semi-quantitative analysis of hypoxia penetrance normalized to the vehicle-control. Horizontal axis labels in **b** also apply to **e**. **c** Hypoxia immunofluorescence imaging in vehicle-control and pharmacological treatment from tadpoles amputated in regenerative (st. 40–41) and refractory (st. 45–46) periods at 1 hpa. Independent experiments gave consistent readouts. Bottom panels: high magnification of correspondent rectangles in middle panels. Green arrowhead: high hypoxia in the wound epithelium (middle panels) and prospective regeneration bud (bottom panels); orange arrowhead: low hypoxia in the prospective regeneration bud (bottom panels); red arrowhead: no hypoxia in the wound epithelium (middle panels) and prospective regeneration bud (bottom panels); white dotted line: posterior tail outline. **d–f** HIF-1α stability levels are significantly affected by depleted ROS and dramatically affected in refractory period. **d** Representative western blot in all conditions at 1 hpa. **e** Semi-quantitative analysis of HIF-1α stability levels normalized to the vehicle-control. **f** HIF-1α immunofluorescence imaging in the regenerative condition at 1 hpa. Independent experiments gave consistent readouts. Yellow arrowhead: high HIF-1α stability in the prospective regeneration bud; white arrowhead: high HIF-1α stability in the wound epithelium. a.u. arbitrary units; scale bars: 100 μm. Statistical analyses were performed by paired Student's t-test (two-tailed p-value). n flows of 20 specimens each, or n blots of 30 specimens each indicated in brackets. NS non-significant; *p < 0.05; **p < 0.01

Hypoxia was then spatially resolved using the same marker by confocal microcopy. The wound epithelium was hypoxic but maximal hypoxia occurred in the prospective regeneration bud. Hypoxia was less penetrant in the DPI-treated tadpoles. In the refractory period, hypoxia was virtually absent in the tails of both vehicle-control and drug-stabilized HIF-1α tadpoles (Fig. 5c).

Next, we investigated the stabilization and localization of HIF-1α in the different conditions at 1 hpa. For a semi-quantitative analysis of HIF-1α levels, tail explants were collected for western blotting[33,34]. DPI-treated tadpoles significantly decreased HIF-1α stability levels by 20% compared with the vehicle-control ($n = 4$ blots of 120 specimens, $p = 0.007$; Fig. 5d, e, Supplementary Fig. 14). DPI decreased both hypoxia and HIF-1α stability, suggesting that ROS might not directly stabilize HIF-1α. To test this, we performed three assays: treated tadpoles with ROS scavenger trolox and determined (i) hypoxia and (ii) HIF-1α stability levels; and (iii) treated tadpoles with NADPH oxidases inhibitor at a dosage higher by one order of magnitude (DPI 10 μM until 1 hpa) and determined HIF-1α stability levels. Treatment with trolox neither affected hypoxia nor HIF-1α levels (Supplementary Fig. 15a–c), and treatment with the higher dose of DPI did not decrease HIF-1α levels proportionately (Supplementary Fig. 15d, e). In the refractory period, HIF-1α levels were dramatically decreased in half (51%), compared with the stage control ($n = 4$ blots of 120 specimens, $p = 0.001$; Fig. 5d, e).

DMOG-treated tadpoles had a 36% non-statistically significant increase in HIF-1α stability compared with vehicle-control st. 45–46 ($n = 4$ blots of 120 specimens, $p = 0.343$; Fig. 5d, e). This effect may probably have been more penetrant and significant if exposure to DMOG were larger than 1 h (other studies typically use 6 or 24 h)[31]. Intriguingly, both uncut and amputated whole tadpoles had drastic differences in HIF-1α stability levels, which were far higher in regenerative than in refractory period tadpoles (Supplementary Fig. 16). These point to an age-dependent stability pattern that might have implications in developmental phenomena beyond regeneration.

HIF-1α was then spatially resolved using confocal microcopy. We showed that HIF-1α was highly stabilized in the wound epithelium and prospective regeneration bud. Importantly, co-localization with the nucleus inferred robust transcriptional activity (Fig. 5f).

Collectively, data demonstrate strong correlations between hypoxia penetrance and HIF-1α stability levels necessary for regeneration. Further, data show that ROS stabilizes HIF-1α indirectly, i.e., via hypoxia.

**HIF-1α does not act downstream of ROS to modulate regeneration.** To further integrate ROS and HIF-1α, we investigated whether ROS acts upstream of HIF-1α to modulate regeneration.

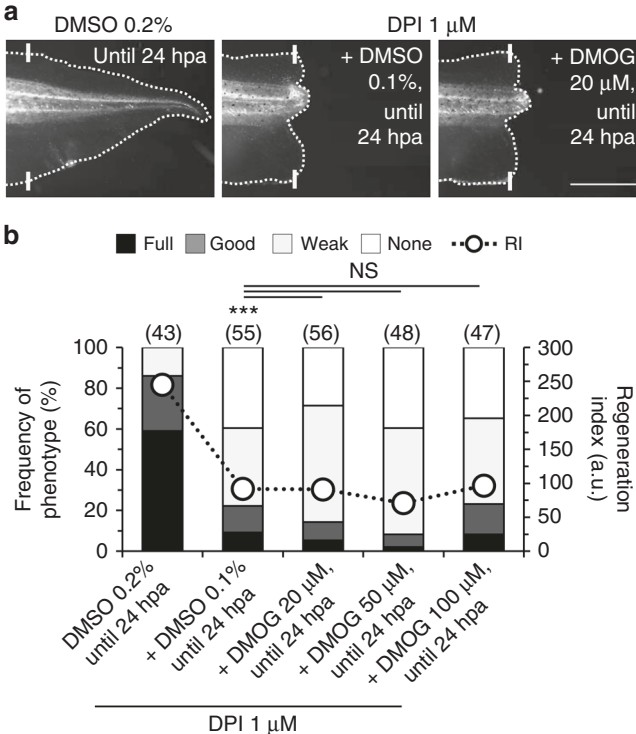

**Fig. 6** ROS do not directly stabilize HIF-1α to modulate regeneration. **a**, **b** Stabilization of HIF-1α does not rescue DPI-impaired regeneration. **a** Representative tails at 7 dpa in vehicle-control and pharmacological treatments from tadpoles amputated at st. 40–41. **b** Qualitative and quantitative analyses of regeneration efficiency for the different conditions tested. White solid lines: amputation plane; scale bar: 1 mm. RI regeneration index; a.u. arbitrary units. Statistical analyses were performed by Fisher's exact test (two-tailed p-value). n biological replicates indicated in brackets. NS non-significant; ***p < 0.001

Using an epistasis assay, we attempted to rescue the DPI-impaired regeneration in the regenerative period by stabilizing HIF-1α with hypoxia-mimicking DMOG. Treatment with DMOG did not rescue DPI-impaired regeneration. The frequency of full and good phenotypes remained unchanged or slightly decreased (DMOG 20 μM: from 9 to 5% for full and from 13 to 9% for good phenotypes). Overall, RI was maintained (DMOG 20 μM: from 92, $n = 55$, to 91, $n = 56$, $p = 0.333$; Fig. 6a, b). Increasing DMOG dose did not improve regeneration (Fig. 6b). Thus, ROS does not directly stabilize HIF-1α to modulate regeneration.

**HSP90 is necessary for regeneration in the same time-window as HIF-1α.** To reveal downstream mechanisms pertaining HIF-1α-mediated regeneration, we searched for candidates that are ideally targeted by HIF-1α and participate in injury recovery. HSP90 is a HIF-1α-induced stress-responsive chaperone that mediates wound healing in vitro and in vivo[14,16,35–37]. Therefore, we tested whether HSP90 is required for regeneration using the well-established HSP90 inhibitor alvespimycin (17-DMAG)[38]. 17-DMAG-treated tadpoles had dramatically abrogated regeneration, with >20-fold decrease in the frequency of full phenotypes (from 44 to 2%) and an increase in the frequency of none phenotypes (from 0 to 54%; RI reduced from 224, $n = 68$, to 60, $n = 48$, $p < 0.0001$; Fig. 7a, b). A refined exposure showed that 17-DMAG until 15 minpa was still equally penetrant with the same frequency of full phenotypes (2%) and similar frequency of none phenotypes (49%; RI reduced from 231, $n = 48$, to 64, $n = 47$, $p <$

0.0001; Fig. 7a, c). The complementary approach showed that 17-DMAG either from 1 or 2 hpa until the end of regeneration (7 days) impaired regeneration in an exposure-independent way. Specifically, treatment with 17-DMAG from 2 hpa decreased the frequency of full phenotypes by 6.4-fold (from 66 to 10%) and increased the frequency of none phenotypes from 0 to 8% (RI reduced from 254, $n = 59$, to 150, $n = 44$, $p < 0.0001$; Fig. 7d). Treatment with 17-DMAG from 3 hpa and beyond did not significantly impair regeneration. From 3 hpa, the frequency of full and good phenotypes were similar (65% and 22%, respectively, negligible fold shift) compared with vehicle-control (RI from 254, $n = 59$, to 249, $n = 43$, $p = 0.418$; Fig. 7a, d).

Together, HSP90 is necessary for regeneration at a precise time-window when HIF-1α is most stable (first 3 hpa). These perfectly parallel results suggest a causal relationship between HIF-1α and HSP90.

**HIF-1α regulates electric current reversal in the regeneration bud.** The above experiments revealed HSP90 as a putative (correlative) downstream target of HIF-1α. To reveal a de facto (causative) one, we next tested whether the important $J_I$ reversal is affected by HIF-1α activity. As expected[8], the majority of vehicle-control bud currents reversed at 6 hpa ($83.33 \pm 9.62\%$) and all at 24 hpa (100%), giving a net inward $J_I$ at both 6 hpa ($-0.10 \pm 0.04\ \mu A\ cm^{-2}$, $n = 16$) and 24 hpa ($-0.15 \pm 0.02\ \mu A\ cm^{-2}$, $n = 17$) (Fig. 7e). Echinomycin-treated tadpoles significantly precluded $J_I$ reversal in the regenerating bud. At 6 hpa, less than half ($45.83 \pm 4.17\%$, $p = 0.014$) reversed, sustaining a significant outward $J_I$ ($2.15 \pm 1.05\ \mu A\ cm^{-2}$, $n = 20$, $p = 0.045$); at 24 hpa, less than a third ($29.17 \pm 11.02\%$, $p < 0.0001$) reversed, sustaining a significant outward $J_I$ ($1.25 \pm 0.64\ \mu A\ cm^{-2}$, $n = 20$, $p = 0.040$) (Fig. 7e).

The spatial profile of $J_I$ showed an increased inward current in echinomycin-treated tadpoles in both spinal cord and dorsal fin, significant at 24 hpa (Supplementary Fig. 18). Together with the outward $J_I$ in the bud, these sustain a dorsoventral circuit loop.

Altogether, the $J_I$ reversal (direction) and $J_I$ magnitude are regulated by HIF-1α, suggesting a mechanism by which HIF-1α modulates regeneration.

## Discussion

Compelling evidence of the pervasive roles of redox activities during regeneration was described recently by our group[8]. Numerous recent studies in widespread regeneration models emphasize the importance of ROS during regeneration[3–7]. However, the systematic integration of key redox state players during regeneration remained elusive. Therefore, in this study, we have attempted to integrate $O_2$, ROS, hypoxia, and HIF-1α cues using a vertebrate regeneration model.

We propose a sequential hypothesis arguing that barrier breaking leads to an $O_2$ influx that fuels local ROS production and together tune the oxic microenvironment in the (prospective) regeneration bud. The ensuing hypoxia is permissive for HIF-1α activity, which in turn modulates regeneration. Our results support the sequences of this hypothesis. First, using an $O_2$ sensing optrode we demonstrated an elevated and steady $O_2$ influx upon amputation ($O_2$ sink) and mapped the spatiotemporal dynamics during regeneration. The profile correlated with the progression of regeneration, presenting a prominent plateau that correlated with bud formation. Next, targeting this plateau (6 and 24 hpa), we demonstrated a higher $O_2$ influx in the refractory period, predicting the regeneration efficiency. The $O_2$ influx defined plateau is regeneration-specific and the $O_2$ sink is an instantaneous and probably conserved response to injury. Inhibition of NADPH oxidases decreased $O_2$ influx, evidencing a causal link between $O_2$ influx and ROS production. The magnitude of $O_2$

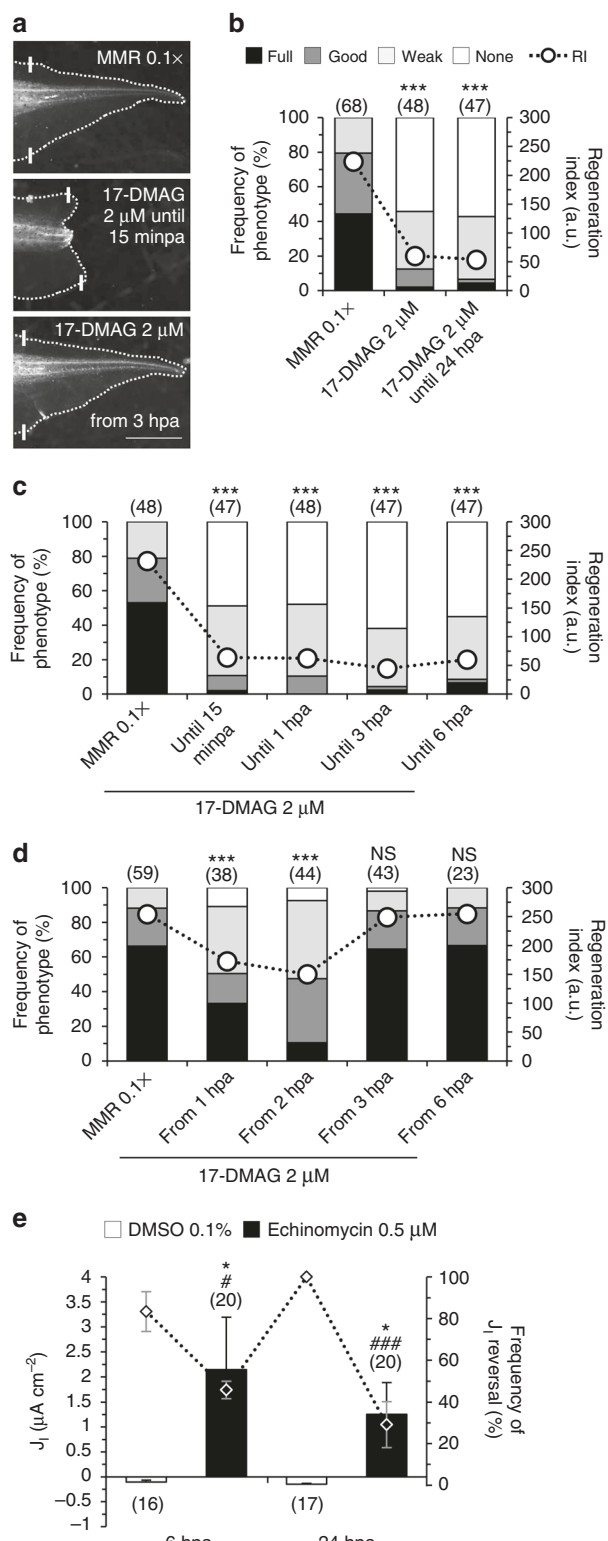

**Fig. 7** HSP90 is a putative and $J_I$ reversal is a de facto downstream targets of HIF-1α. **a–d** Loss of regeneration by HSP90 inhibition and temporal requirement for HSP90 activity during regeneration. **a** Representative tails at 7 dpa in control and pharmacological treatment from tadpoles amputated at st. 40–41. **b–d** Qualitative and quantitative analyses of regeneration efficiency for the different conditions tested. White solid lines: amputation plane; scale bar: 1mm. RI regeneration index; a.u. arbitrary units. Stacked bars legend in **b** applies to **c** and **d**. **e** HIF-1α regulates the $J_I$ reversal hallmark. $J_I$ in regeneration bud measured in two different time-points in vehicle-control and echinomycin-treated tadpoles amputated at st. 40–41. Statistical analyses were performed by Fisher's exact test **b–d**; **e** #, vs. $J_I$ reversals), or unpaired Student's t-test (both two-tailed p-value) (**e**: *, vs. $J_I$ magnitude). The data are presented as mean ± s.e.m. n biological replicates indicated in brackets. NS non-significant; */#p < 0.05; ***/###p < 0.001

An epistasis assay confirmed that HIF-1α did not act downstream of ROS to modulate regeneration. Finally, we analyzed candidate downstream targets of HIF-1α. HSP90 was found to be necessary for regeneration in the same time-window as HIF-1α, and HIF-1α activity was required for electric current reversal. Together, these suggested HSP90 as a putative (correlative) and $J_I$ reversal as a de facto (causative) downstream targets of HIF-1α to modulate regeneration. Altogether, the results provide evidence for an instantaneous injury-induced $O_2$ sink that fuels required ROS production and together stabilize HIF-1α by hypoxia to modulate regeneration possibly via HSP90 and $J_I$ reversal. This sequence of events is postulated as a mechanism of action (Fig. 8, Supplementary Fig. 19).

Intriguingly, while recording, many $O_2$ flux measurements appear to stabilize in an oscillatory rather than in a flat plateau. $O_2$ oscillations with a period ranging between ~2 and 4 min (frequency between ~4 and 8 mHz) were relatively common (Supplementary Fig. 6). This study was not designed to address $O_2$ oscillations (which likely require longer recordings, e.g., of 30 min); nonetheless, they might be an epiphenomenon of oscillations in metabolism (e.g., mitochondrial activity) or signaling (e.g., $Ca^{2+}$), and/or might be an intrinsic signaling cue[39,40]. That being said, NAD(P)H metabolite concentration is oscillatory[41]. NADPH donates an electron to $O_2$ to generate a ROS; therefore, it is feasible to speculate that $O_2$ oscillations might be, at least in part, the result of oscillatory $O_2$ oxidation, since exogenous $O_2$ fuels ROS. The roles of $O_2$ oscillations in general and during regeneration in particular warrant future work (Supplementary Fig. 6)[42,43].

The existence of $O_2$-specific plasmalemmal translocators (passive or active) is unknown. Aquaporin-1 has been identified as a non-selective translocator of $O_2$[44]; however, anatomic and physiologic intricacies such as the rete mirabile in fish swim bladder, point to no active (against gradient) $O_2$ transmembrane transporters. Thus, with or without a barrier, the $O_2$ flux follows passive diffusion in a supply-demand way. Injury disrupts local vasculature and together with infection, inflammation, oxidative burst, cell migration, and cell proliferation, comprise the factors leading to chronic wound hypoxia[45,46]. This established rationale may lead to a paradox where large amounts of $O_2$ are being consumed in multiple fronts without an efficient blood supply. In wound and regeneration models, ROS have been shown to precede oxidative burst; in fact, ROS attract the immune cells responsible for the oxidative burst[6,47,48]. Further, ROS has been consistently demonstrated as essential for regeneration[3–8]. The same studies showed that ROS production is a steady and long-lasting response to injury, creating an additional burden for $O_2$ demand, further highlighting the paradox. Therefore, we

influx was independent of ROS per se, excluding a feedback loop between reagent ($O_2$) and product (ROS). Inhibition of HIF-1α dramatically abolished regeneration, while its stabilization induced regeneration in the refractory period. Hypoxia correlated with HIF-1α stability levels, whose co-localization in the prospective bud underlies regeneration efficiency. Experiments modulating ROS production and scavenging demonstrated that ROS did not stabilize HIF-1α directly but indirectly via hypoxia.

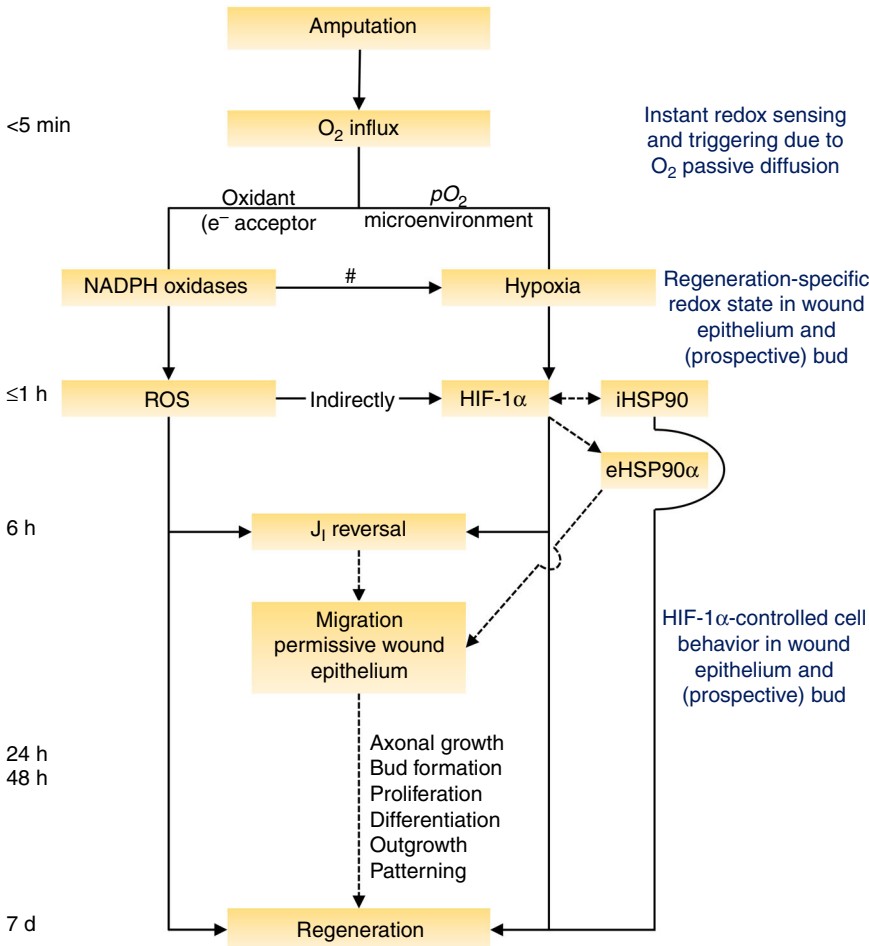

**Fig. 8** Stepwise model integrating redox state activities during regeneration. Instantaneous injury-induced $O_2$ influx fuels ROS production and together tune a permissive $pO_2$ microenvironment—hypoxia—for HIF-1α stabilization. Early time-window up to 6 hpa is focused, because is when the magnitude of $O_2$ influx correlates with regeneration and when required ROS and HIF-1α activities occur. ROS per se do not feedback with the magnitude of $O_2$ influx. ROS do not directly stabilize HIF-1α but do so indirectly by regulating hypoxia in the bud owing to local $O_2$ consumption and $O_2$ influx demand (Supplementary Fig. 19). We infer that intracellular HSP90 is at least partially required for early hypoxia-induced stabilization of HIF-1α, resulting in the secretion of eHSP90α. 6 hpa is also the time-point of the hallmark $J_I$ reversal, an accurate predictor of regeneration efficiency that mediates redox-modulated regeneration. HIF-1α regulates $J_I$ reversal, pointing to an integration of HIF-1α with the bioelectric state, in series or in parallel with ROS. Mechanistically, HIF-1α modulates regeneration via HSP90/eHSP90α and $J_I$ reversal presumptive effect on cell migration migration to form the wound epithelium and/or regeneration bud. iHSP90: intracellular HSP90; eHSP90α: extracellular HSP90α. Solid line arrows: demonstrated; dotted line arrows: hypothesized/probable. # mechanism schematized in Supplementary Fig. 19; * demonstrated in ref. [8]

theorized that exogenous $O_2$ fuels local ROS production in the regeneration bud. Indeed, using reverse analysis methodology, we showed that at least half to two-thirds of the required ROS production in regenerative tadpoles depends on influx of exogenous $O_2$ into the bud. Instant ROS production might prevent an acute hyperoxia, which would derive from the observed instant and steady $O_2$ influx. Nonetheless, some level of acute hyperoxia or injury oxidation is not entirely excluded, and it could, for example, contribute to localized cell death. The use of an antioxidant proved that production, not ROS *per se*, decreased $O_2$ influx causally. Moreover, unaltered $O_2$ influx in antioxidant-treated tadpoles showed that ROS do not feedback with the magnitude of $O_2$ influx, suggesting that NADPH oxidases activity is the major driving force for $O_2$ flux. Other motive forces (cumulative) are likely to be proliferation and bud maturation at 24 hpa, since a peak in $O_2$ influx was noted at this time-point. Our data does not entirely exclude the possibility of $O_2$-induced ROS production as previously observed[49]. Thus, it is likely that exogenous $O_2$ modulates regeneration by fueling required

ROS production, making $O_2$ a potential target to enhance regeneration, in addition to its routine applications in chronic wounds[50–52].

The higher $O_2$ influx in the refractory period was independent of its deficient ROS production. This raises two important questions: how the higher $O_2$ influx mechanistically abolishes regeneration? and, why an elevated $O_2$ influx occurs during the refractory period when ROS production is lower? For the first question, we postulated that a higher $O_2$ influx leads to a non-hypoxic bud. This non-permissive oxic microenvironment degrades HIF-1α, subsequently abolishing regeneration. Our data on the quantification and imaging of hypoxia and HIF-1α substantiate this rationale. For the second question, we inferred that depleted ROS production and negligible proliferation and bud maturation, point to less demand for $O_2$. Nurturing the paradox, refractory period tails have a wound epithelium thicker than regenerative tadpoles[24,53,54] and hence may be less permeable to $O_2$. Size does not explain the robust influx though. Altogether, the driving causes for the regeneration-dependent higher $O_2$ influx

remain elusive and require dedicated research in the future. Differential mitochondria activity might be a prime hypothesis to test.

Inhibition of HIF-1α abolished regeneration when the inhibitor was present during the first 3 hpa. Concurrently, HSP90 activity was required exactly during that time-window. These acute requirements put HIF-1α and HSP90, together with ROS[8], among the earliest and most pervasive biochemical limiting factors of regeneration. Drug-stabilized HIF-1α-induced regeneration. The results on HIF-1α modulation are in agreement with a recent study demonstrating that HIF-1α is necessary for and sufficient to induce regeneration in mouse ear hole[18]. In addition, our results show that long-term (chronic) stabilization of HIF-1α does not adversely affect regular regeneration. Future studies are required to test whether other HIF transcription factors (especially HIF-2α) play a role in regeneration.

It is well known that hypoxia stabilizes HIF-1α[9] and that mesenchymal condensations, such as limb bud and tumors[13,20], are hypoxic. We thus inferred that the regeneration bud would be hypoxic. Indeed, we observed strong hypoxia in regenerating conditions but not in the refractory period. This differential penetrance in hypoxia predicted regeneration and HIF-1α stabilization. Congruently, hypoxia matched HIF-1α stability and localization, and correlated with regeneration. ROS have been suggested to stabilize HIF-1α in hypoxic and non-hypoxic conditions[55]. Whether ROS, in particular $H_2O_2$, can directly stabilize HIF-1α via inhibition of PHD enzymes has been a contentious field of research[56,57]. Another study showed activation of HIF-1α promoter by ROS[58]. In our study, ROS production had a substantial effect on HIF-1α stability; however, five lines of evidence suggest that this effect was not due to direct HIF-1α stabilization by ROS: (i) HIF-1α stability was independent of NADPH oxidases inhibitor dosage; (ii) NADPH oxidases inhibition proportionally decreased hypoxia and HIF-1α levels (41% reduction in hypoxia translated to a 20% decrease in HIF-1α levels), which were remarkably identical to those observed in refractory period tails (70% reduction in hypoxia translated to a 51% decrease in HIF-1α levels); (iii and iv) ROS scavenging—which did not affect $O_2$ influx—neither changed hypoxia nor HIF-1α levels; and (v) ROS did not activate HIF-1α to modulate regeneration as determined by an epistasis assay. Together, these data demonstrated that ROS indirectly stabilized HIF-1α via hypoxia. In regenerating conditions, the balance between $O_2$ influx, $O_2$ consumption and ROS production resulted in bud hypoxia to stabilize HIF-1α. Suppression of ROS production nevertheless exhibited considerable hypoxia (59% of control) and consequent HIF-1α stability (80% of control) that is due to the decreased $O_2$ influx. Therefore, the effect on hypoxia/HIF-1α might not be sufficient to fully account for the robust impairment of regeneration by depletion of ROS alone (Fig. 8, Supplementary Fig. 19)[8].

Simultaneous temporal requirement for HSP90 and HIF-1α activities lead us to speculate that intracellular HSP90 acts alongside while its extracellularly secreted form (eHSP90α) acts downstream of HIF-1α (Fig. 8). Substantiating this, we showed a time-dependent relationship between HSP90 and HIF-1α in regeneration, where early inhibition of HSP90 or HIF-1α abolished regeneration, but late inhibition had no effect. This suggests that HSP90 is important, at least partially, for hypoxia-induced HIF-1α activity. This is consistent with previous studies demonstrating that cytoplasmic binding of HIF-1α to HSP90 is oxygen-dependent: weaker in hypoxia but robust under high oxygen levels. The binding prevents nuclear shuttling of cytoplasmic HIF-1α[59,60]. In mammalian cells in vitro, stabilization of HIF-1α in hypoxia was precluded by inhibition of HSP90[59]. Further, HIF-1α induced the secretion of eHSP90α that extracellularly mediates fibroblast migration in mammalian wound

healing in vivo[61]. Congruently, HIF-1α-induced HSP90 was shown to induce fibroblast migration in scratch wounds in vitro[14]. Collectively, we thus infer that intracellular HSP90 is at least partially required for hypoxia-induced stabilization of HIF-1α, resulting in secretion of eHSP90α to promote cell migration to form the wound epithelium and/or regeneration bud (Fig. 8).

Other studies demonstrate the importance of HIF-1α in cell behaviors essential for regeneration. HIF-1α regulates chemotaxis of neural crest in *Xenopus* and of progenitor cells to the wound in mouse ischemic model[34,62]. Additionally, HIF-1α regulates epithelial-to-mesenchymal transition and targets pluripotency regulators, such as Oct3/4, Nanog, Notch, and Wnt[11,18,34]. Therefore, control of the cell pool identity and stemness are other candidates, although not mutually exclusive, to explain the effect of HIF-1α in regeneration. Several canonical signaling pathways, such as Wnt, FGF, and BMP, are known to be required for tail regeneration[24]. How and when hypoxia/HIF-1α/HSP90 concertedly interact with the various signaling pathways to mediate regeneration remains to be elucidated.

Regeneration is undoubtedly a complex multifactorial phenomenon. Integrating HIF-1α with bioelectric activities that crosstalk with redox activities[8] can provide a more comprehensive and holistic understanding of the roles they play in regeneration. Excitingly, we demonstrated that HIF-1α is required for $J_I$ reversal, a potential hallmark of regeneration[8,54]. Without reversal, a cathode is maintained in the amputation edge that might contribute to cell overmigration. Previously, we showed that $H_2O_2$ switches $J_I$ reversal in the bud[8]. HIF-1α may then act in series or in parallel with $H_2O_2$ to modulate $J_I$. Among other effects, electric currents and consequent electric fields (EF) guide cell migration[63], reinforcing this cell behavior as a prime candidate mechanism underlying HIF-1α-modulated regeneration (Fig. 8). Indeed, a recent study showed that hypoxic preconditioning enhanced EF-induced keratinocyte migration and wound healing[64].

In conclusion, this study integrates the roles of $O_2$, ROS, and HIF-1α cues during regeneration, deepening the understanding of the redox activities during this demanding process. Instantaneous exogenous $O_2$ influx fuels required ROS production and together stabilize HIF-1α by hypoxia that, ultimately, modulates regeneration possibly via HSP90 and $J_I$ reversal. Redox state players and dynamics might thus reserve important targets for translational medicine.

## Methods

**Tadpoles**. Animal procedures and euthanasia were approved by the Marine Biological Laboratory (MBL; Woods Hole) (protocol no. 14-59) and the University of California, Davis Institutional Animal Care and Use Committees (protocol nos 18601 and 20337). *Xenopus laevis* (Daudin, 1802) tadpoles were acquired from National Xenopus Resource (NXR; www.mbl.edu/xenopus) or Xenopus Express (www.xenopus.com), arriving in batches of 50–400 animals. For data accuracy, we decided not to mix animal sources. Thus, tadpoles from NXR were used to generate the data shown in Figs. 1b, 2c, Supplementary Figs. 1, 3a, 6a–d; all other data were generated with tadpoles from Xenopus Express. Sorted and staged[25] tadpoles were transferred to fresh Marc's modified Ringer (MMR) 0.1× medium composed of (mM): NaCl 10, CaCl₂·2H₂O 0.2, KCl 0.2, MgCl₂·6H₂O 0.1, and HEPES 0.5 (pH 7.1–7.2) (Sigma-Aldrich). Tadpoles were incubated between 13 and 33 °C until reached the correct stages for amputation, st. 40–41 (regenerative) or st. 45–46 (refractory period).

**Oocytes**. Pre-sorted *X. laevis* oocytes were acquired from Xenopus 1 (www.xenopus1.com). Upon arrival, oocytes were transferred to MMR 1× (NaCl 100 mM, CaCl₂·2H₂O 2 mM, KCl 2 mM, MgCl₂·6H₂O 1 mM, and HEPES 5 mM; pH 7.1–7.2 (Sigma-Aldrich)) and stored overnight at 13 °C prior to experiments.

**Mice**. Eight-week-old male *Mus musculus* (Linnaeus, 1758) BKS.Cg-*Dock7ᵐ* +/+ *Leprᵈᵇ*/J heterozygous (non-diabetic) mice were acquired from The Jackson Laboratory (www.jax.org) for an unrelated study[65]. Mice euthanasia was approved

by local (University of California, Davis) Institutional Animal Care and Use Committee (protocol no. 16766).

**Tail regeneration assay**. Half of the tail of randomized normal tadpoles, equilibrated to room temperature and immobilized in myosin inhibitor N-benzyl-p-toluene sulfonamide (BTS; Tocris Bioscience, cat. no. 1870) 50 μM, was amputated with a scalpel (blade no. 10; Feather Safety Razor) (Fig. 1a). Experimental conditions were refreshed daily and tail photomicrographs were taken at 7 days post-amputation (dpa). Regeneration efficiency was scored using the regeneration index (RI), calculated from the frequencies (f) of the outcome predefined phenotypes using the following equation[23,53]:

$$RI = (f_{Full} \times 3) + (f_{Good} \times 2) + (f_{Weak} \times 1) + (f_{None} \times 0), \qquad (1)$$

RI ranges from 0 (if all none) to 300 (if all full), in arbitrary units. For guidance, when RI is ≥250 it represents virtually unimpaired regeneration; and, when RI is ≤100 it represents drastically impaired regeneration. Over length or area, RI provides an advantageous measure of the morphogenetic quality (axes outgrowth and patterning) of regenerated tails.

**Fin wound healing assay**. Randomized normal tadpoles (st. 40–41) equilibrated to room temperature and immobilized were wounded using a 2 mm biopsy punch (Miltex, cat. no. 33–31) that removed a rounded portion of the dorsal fin (at A/P axis intersection; Supplementary Fig. 4a,b). Healing was followed and photomicrographed as in the amputated tails. Wound area was measured using ImageJ (http://rsbweb.nih.gov/ij/) and treated using Excel (Microsoft).

**Oocytes wounding**. Oocytes equilibrated to room temperature were wounded using a heat-pulled glass capillary with a broken tip (diameter of ~75 μm) mounted on a manual micromanipulator and impaled through the oocyte membrane[26]. Wounds were made around the center of the animal pole. A nylon mesh (800 μm pore size; nitex mesh) glued to the Petri dish provided support and immobilization to the oocytes during capillary impalement and optrode measurements (Supplementary Fig. 5a).

**Mice skin wounding**. Killed mice were kindly dispensed by Yunyun Shen (UC Davis; Department of Occupational and Environmental Health, Zhejiang University, China), after eyes removal for unrelated purposes[65]. A square patch of hair from the back (dorsal axis) was removed with hair remover cream (Nair). Naked skin was cleaned with ethanol 75% and washed with deionized water prior to wounding. A full thickness skin wound of <1 cm length was made at around the L/R and A/P axes intersection using a scalpel (blade no. 15; Henry Schein) (Supplementary Fig. 5d).

**Pharmacologic modulations**. DPI (Sigma-Aldrich, cat. no. D2926) 1 and 10 mM, trolox (Cayman Chemical, cat. no. 10011659) 80 mM, echinomycin (Cayman Chemical, cat. no. 11049) 0.5 mM, chetomin (Cayman Chemical, cat. no. 14437) 0.5 mM, and DMOG (Cayman Chemical, cat. no. 71210) 20 and 100 mM were stocked in dimethyl sulfoxide (DMSO; Sigma-Aldrich, cat. no. D2650) at −20 °C. 17-DMAG (Selleckchem, cat. no. S1142) 765 μM was stocked in phosphate-buffered saline (PBS; HyClone, cat. no. SH30264) at −20 °C. Working solutions—drugs and vehicle (DMSO) reconstituted in MMR 0.1×—were freshly prepared prior to application via immersion (bath), in the doses and exposures specified. Extensive randomized dose-exposure screenings were designed for a final DMSO concentration of typically ≤0.1% (Supplementary Figs. 8, 10–12, 17). For delimited exposures, drugs were washed in and to the respective control (MMR 0.1×) or vehicle-control (DMSO 0.1%). Tadpoles treated with the light-sensitive drug 17-DMAG were followed in the dark. Dosage and exposure used did not meaningfully affect development or mortality (Supplementary Figs. 8, 10–12, 17). However, long-term exposure (throughout regeneration) to 17-DMAG resulted in increased mortality after 6 dpa. Therefore, tail photomicrographs were taken at 6 dpa, exclusively for this case. This change does not compromise data reliability, because tails at 6 or 7 dpa have the regeneration phenotype equally well defined. Matched sibling controls were performed for every drug treatment used in all readouts.

**Optrode measurement**. Extracellular net dissolved O2 flux ($J_{O_2}$) was measured non-invasively with a self-referencing O2-selective optrode. This probe is a pulled optical fiber with a solid state O2-sensitive fluorophore coating in the tip. O2 is quantified by fluorescence quenching after excitation of the fluorophore with blue–green light ($\lambda = 505$ nm) from a LED source[66]. System has high spatial (~20–50 μm) and temporal (~2 s) resolutions. Unlike polarographic electrodes, optrode does not consume O2 in the measurement and has the capacity to measure in the gas phase. Our measurements were performed always in the liquid phase (experimental condition specified). Ready-to-use needle-type housing optrodes (PreSens, NTH-PSt1-L5-TS-NS40) were incorporated into the turn-key system scanning micro-optrode technique (SMOT; Applicable Electronics).

Prior to measurements, a two-point calibration of the optrode was performed in 0 and 20.95% pO2 solutions. 0% pO2 was achieved with saturated sodium bisulfite (mixture of NaHSO3 and Na2S2O5; Sigma-Aldrich, cat. no. 243973) 2 M and 20.95% pO2 was achieved with bubbled deionized water; for the bubbling was used an aquarium pump to push atmospheric air through an airstone during 20 min. Tadpoles were immobilized in myosin inhibitor (BTS 50 μM) throughout measurements, except to test the effect of chemical immobilization on O2 flux (Supplementary Fig. 1d). Under the microscope, immobilized tadpoles and optrode were positioned in the measuring chamber half filled with MMR 0.1× not supplemented or not with vehicle-control/drugs. During measurements, optrode was as close as possible from tail surface (~10 μm) and excursed 30 μm away this position (far pole) and then back (near pole; 11 s per iteration or ~0.1 Hz). Reference values were recorded with optrode away from tail (>>1 mm) (Supplementary Fig. 1a–c). $J_{O_2}$ was recorded for 2–5 min (~10–30 data points), usually sufficient for a consistent signal to be averaged. Measurements were performed at room temperature in the regions and times specified. Data and metadata were acquired and extracted using ASET-LV4 (Science Wares) and treated using Excel (Microsoft).

O2 flux measurements and data acquisition, extraction, and treatment were performed in fins, oocytes, and mice skin (Supplementary Figs. 4a, 5a,d) as in the amputated tails with minor changes. In oocytes, wounded measurements were acquired from ~2 to ~15 minpw. This period was averaged to give a single value per specimen to pair with the intact measurement. In mice skin, the measuring chamber was a 100 mm Petri dish filled with PBS (AMRESCO, cat. no. E404). Intact (0 minpw) measurements were acquired at 33.7 ± 4.7 min post-mortem; wounded measurements were acquired from ~10 to ~30 minpw. This period was averaged to give a single value per specimen to pair with the intact measurement.

The ASET software communicates with the electronic firmware that has an embedded microprocessor that automatically and in real-time calculates pO2 (%), using the fluorescence lifetime-based method. This method follows the Stern-Volmer equation based on a measurement of phase angle shift[66,67] relative to calibration values obtained (at known temperature and pressure) for the sensor being used. From these extractable raw data ($pO_2$), O2 concentrations ([O2]) were calculated using the following adapted[67] equation:

$$[O_2](\mu M) = \frac{p_{atm} - p_W(T)}{p_N} \times \frac{\frac{pO_2}{0.2095}}{100} \times 0.2095 \times \alpha(T) \times 1000 \times \frac{1}{V_M}, \qquad (2)$$

where $p_{atm}$ is the atmospheric pressure (1013.25 mbar at sea level), $p_W(T)$ is the vapor pressure of water (26.507 mbar at 22 °C (mean room temperature)), $p_N$ is the standard atmospheric pressure (1013.25 mbar), $\frac{\frac{pO_2}{0.2095}}{100}$ is the ratio of O2 in the gas mixture (referred elsewhere[67] as Q), $\alpha(T)$ is the Bunsen absorption coefficient (29.908 cm³(O2) cm⁻³ at 22 °C) and $V_M$ is the molar volume (22.414 l mol⁻¹). Some of these parameters were obtained from further calculations and/or standard curves/tables, consulted in the optrode manufacturer's instruction manual. The [O2] was then converted to pmol cm⁻³ and included in the Fick's first law to calculate the fluxes:

$$J_{O_2}(pmol\, cm^{-2}\, s^{-1}) = -D \times \frac{\delta O_2}{\delta x}, \qquad (3)$$

where D is the diffusion coefficient of dissolved O2 ($2.42 \times 10^{-5}$ cm² s⁻¹) and $\delta O_2$ is the concentration difference (in pmol cm⁻³) over the excursion $\delta x$ (30 μm). Finally, reference mean value was subtracted to each sample flux data point and then sample flux was averaged. Negative values mean net influx (O2 entering the animal tissue) and positive values mean net efflux (O2 exiting the animal tissue).

Optrode measurements were acquired using two SMOT systems: one mounted at MBL and a similar one (most parts from MBL system) mounted at University of California, Davis. For data accuracy, we decided not to mix the measurements from the different SMOT systems. Thus, the data presented in Figs. 1b, 2c, Supplementary Fig. 1c–e, 3a, 6a–d were acquired using the SMOT system at MBL; all other data were acquired using the other SMOT system. Mean room temperature in MBL was 23 °C; therefore, temperature-dependent parameters of Eq. 2 were readjusted accordingly.

**Gills and cutaneous respiration**. Surface O2 uptake measurements (Supplementary Fig. 2a) and data acquisition, extraction, and treatment were performed as in the amputated tails.

**H2O2 fluorescence imaging**. H2O2 was imaged and semi-quantified using the transgenic X. laevis HyPer line (Xla.Tg(Hsa.UBC-Gal4;UAS:HyPer-YFP)^Amaya) obtained from National Xenopus Resource (cat. no. NXR_0.0127)[6,68,69]. Individually immobilized tadpoles at the specified condition were placed in a 2 well lamina and covered with a lamella. A UPLSAPO ×10/NA 0.40 objective (Olympus) mounted in an inverted confocal microscope (Olympus FV1000 confocal system) was used for the imaging. Tails were z scanned at 405 (Alexa Fluor 405 channel) and 488 (Alexa Fluor 488 channel) nm excitations (ex) and detected at 515 nm emission (em; both Alexa Fluor channels' settings were edited to change their default emission maxima to 515 nm). Scanning was sequential (first 405/515 ex/em nm, then 488/515 ex/em nm). Maximum z projections were pseudo-colored in gray

and mean pixel intensity was measured from a circular ROI placed in the (prospective) regeneration bud and in the background (annotated in Supplementary Fig. 7a). After background subtraction, a final pixel intensity value was achieved for both 405/515 and 488/515 nm spectra. The values obtained with ex/em at 405/515 nm were always very low (thus considered as background fluorescence) and did not shift across conditions (ROS/$H_2O_2$/amputation); although fluorescence intensities varied markedly across conditions in the 488/515 nm ex/em spectra. Thus, a ratio of intensities at [488/515 nm] to [405/515 nm] would have resulted in very large values that may falsely be perceived artefactual. Therefore, for clarity, we only present the values for fluorescence obtained with ex/em at 488/515 nm and considered the fluorescence values in the ex/em at 405/515 nm spectra as background intensities in this study. The acquisition settings were kept constant across experiments to allow for ready cross comparison. Technical negative controls (for autofluorescence and wavelength selectivity), transgenesis' negative control (i.e., no HyPer expression) and experimental positive control (high $H_2O_2$) gave conservative expected readouts (Supplementary Fig. 7a). Data and metadata were acquired and extracted using FluoView (Olympus) and treated using ImageJ and Excel.

**ROS fluorescence imaging**. ROS was imaged and semi-quantified using the vital dye chloromethyl derivative of 2',7'-dichlorofluorescein (CM-$H_2$DCFDA; Molecular Probes; cat. no. C6827) in wild-type tadpoles. Stock dye was freshly prepared in anhydrous DMSO (Sigma-Aldrich, cat. no. 276855) at 1 mM. Tadpoles were incubated at room temperature in the dark in fresh CM-$H_2$DCFDA 10 μM for 1 h. Dye was washed out prior to fluorescence imaging. Individually immobilized tadpoles at the specified condition were placed in a small Petri dish under a fluorescence upright microscope (Zeiss Lumar V12) with attached monochromatic CCD camera (Zeiss AxioCam MRm). Tails were imaged using the GFP (488 nm) filter set channeling the light from a metal halide lamp of a fluorescence illumination system (EXFO X-Cite 120). Fluorescent images were pseudo-colored in gray and mean pixel intensity was measured from a rectangular ROI englobing the whole-imaged tail and a circular ROI in the background (Supplementary Fig. 7d). After background subtraction, a final pixel intensity value was achieved for the GFP channel. The acquisition settings were kept constant across experiments to allow for ready cross comparison. Technical negative controls (for autofluorescence and wavelength selectivity) and experimental positive control (high $H_2O_2$) gave conservative expected readouts (Supplementary Fig. 7d). Data and metadata were acquired and extracted using AxioVision software (Zeiss) and treated using ImageJ and Excel.

**Flow cytometry**. Hypoxia was semi-quantified by flow cytometry (fluorescence-activated cell sorting, FACS) of dissociated cells from tail explants[70] using the marker pimonidazole (Hypoxyprobe, cat, no. HP2-100Kit). In hypoxia or $pO_2$ less than 1.1% (8 mmHg or 11 hPa) at 22 °C (mean room temperature), pimonidazole is reductively activated, forming stable adducts with thiol-containing proteins where antibody binds. Tadpoles were incubated in pimonidazole HCl 300 μM for 1 h and, per condition, 20 tail explants (0.5 mm proximal to amputation plane) were incubated in 1 ml of dissociation solution composed of DNase I 100 U ml$^{-1}$ (Roche, cat. no. 4716728001) and Liberase 0.25 mg ml$^{-1}$ (Roche, cat. no. 5401119001) (in PBS), at 28 °C for ~30 min. Cells were washed (FACS buffer: bovine serum albumin (BSA) 1% in PBS) and live/dead stained (Aqua stain; Life Technology, cat no. L34965) at room temperature for 30 min. After fixation (formaldehyde 3.7% in PBS), cells were incubated in mouse IgG$_1$ anti-pimonidazole fluorescein (FITC)-conjugated monoclonal antibody 1:50 (in FACS buffer) at room temperature for 1 h. Cells were passed through a 35 μm cell strainer (Corning, 352235) and 20,000 to 25,000 cells were analyzed on a BD LSRFortessa flow cytometer (BD Biosciences). Geometric mean fluorescence intensities (MFI) were calculated per condition after gating live single cells (Supplementary Fig. 13a–c). Across experiments, MFI were standardized (Supplementary Fig. 13h, j) using the equation:

$$x_{FMO}(MFI) = \frac{x(MFI) - FMO(MFI)}{FMO(SD)}, \qquad (4)$$

where $x_{FMO}(MFI)$ is the standardized geometric mean fluorescence intensity of hypoxia per experiment, $x(MFI)$ is the geometric mean fluorescence intensity of hypoxia per experiment, $FMO(MFI)$ is the geometric mean fluorescence intensity of the fluorescence minus one (FMO) per experiment and $FMO(SD)$ is the standard deviation of the FMO per experiment. Standardization method was then applied per condition to obtain final geometric MFI value. Data were acquired using FACSuite (BD Biosciences) and treated using FlowJo (FlowJo).

**Immunohistochemistry**. Hypoxia was imaged by immunofluorescence of marker pimonidazole in cryosections, as recommended by marker's manufacturer. Tadpoles were incubated in pimonidazole HCl 300 μM for 1 h, fixed in paraformaldehyde (PFA) 4% at 4 °C for 2 h, dehydrated in sucrose 30% (in PBS) at 4 °C overnight and embedded in optimal cutting temperature (OCT) compound for sectioning in a cryotome. Slides with 5 μm sections were rehydrated and permeabilized (Tris-buffered saline with Tween 20 0.1%, TBST) for 10 min, blocked (goat serum 10% in PBS, plus Tween 20 0.1%) at 37 °C for 1 h and incubated in

mouse IgG$_1$ anti-pimonidazole FITC-conjugated monoclonal antibody 1:50 (in blocking solution) at room temperature for 1 h. After washings (TBST), slides were incubated with DAPI (Novus Biologicals, cat. no. NBP2-31156) 1:1000 (in PBS) for 5 min and mounted (Fluoromount-G; SouthernBiotech, cat. no. 0100-01).

HIF-1α was spatially imaged by immunofluorescence in cryosections. Slides were obtained and treated as above with a difference in the blocking solution (goat serum 10% in PBS, plus Triton X-100 0.3%). Slides were incubated in rabbit anti-HIF-1α primary polyclonal antibody (Abcam, cat. no. ab2185) 1:500 (in blocking solution) at 4 °C overnight. After washings (TBST), slides were incubated in goat anti-rabbit IgG (H + L) Alexa Fluor 488-conjugated secondary antibody (Jackson ImmunoResearch Laboratories, cat. no. 111-545-144) 1:200 (in blocking solution) at room temperature for 1 h, stained with DAPI and mounted.

Slides were imaged using UPLSAPO 20×/NA 0.75 and PLAPO 40×WLSM/NA 0.90 (water) objectives (Olympus) mounted in an inverted Olympus FV1000 confocal microscope. Excitation wavelengths were 405 nm for DAPI and 488 nm for FITC/Alexa Fluor 488. The imaging settings were kept constant across conditions and experiments to allow reliable cross comparison. Control slides without antibodies had conservative readouts; however, owing to autoflorescence in the 488 nm excitation, up to the first quarter of the fluorescence intensity signal was excluded. Several independent experiments were performed. Data and metadata were acquired and extracted using FluoView (Olympus) and treated using ImageJ (http://rsbweb.nih.gov/ij/) and Excel.

**Western blotting**. HIF-1α stability levels were semi-quantified by western blotting[33,34]. Per condition, 30 tail explants (0.5 mm proximal to amputation plane) were incubated for 30 min on ice and homogenized in 75 μl of cold lysis solution composed of RIPA buffer (Thermo Scientific, cat. no. 89900) with a protease inhibitor mix (Halt cocktail 3× and EDTA 1× (Thermo Scientific, cat. no. 78430), calpain inhibitor I 260 μM (Cayman Chemical, cat. no. 14921) and MG-132 5 μM (Selleckchem, cat. no. S2619)). For whole tadpoles' analysis, we used 5 tadpoles lysed in 100 μl lysis solution. Samples (40 μg; DC protein assay; Bio-Rad, cat. no. 5000111) were loaded into 4–12% Novex gels (Invitrogen, cat. no. XP04120) for electrophoresis. Following transfer, nitrocellulose membranes (Invitrogen, cat. no. LC2001) were washed (TBST), blocked (fetal bovine serum 10%, SuperBlock 10% (Thermo Scientific, cat. no. 37515), fish gelatin 3% (VWR, cat. no. M319), sodium azide 0.02%, in PBS, pH 7.4) at 37 °C for 1 h and incubated in rabbit anti-HIF-1α polyclonal/mouse IgG$_1$ anti-actin monoclonal primary antibodies (Abcam, cat. no. ab2185/Invitrogen, cat. no. MA5-11869) 1:250/1:2000 (in blocking solution) at 4 °C overnight. Membranes were incubated in goat anti-rabbit IgG (H+L)/goat anti-mouse IgG (H+L) horseradish peroxidase-conjugated secondary antibodies (Abcam, cat. no. ab205718/KPL, cat. no. 474-1806) 1:1000/1:10000 (in blocking solution) at 37 °C for 1 h, washed and detected by enhanced chemiluminescence (ECL) reaction (Advansta, cat. no. K-12045). Nuclear HIF-1α lysate (5 μg; Abcam, cat. no. ab180880) was used to validate the selected anti-HIF-1α antibody. Data were acquired and extracted using VisionWorks (UVP) and treated using ImageJ and Excel.

**Vibrating probe measurement**. Extracellular net electric current density ($J_I$) was measured non-invasively with a vibrating probe[8]. Prior to measurements, a platinum-electroplated probe (~30 μm ball diameter) vibrating at 100–200 Hz was calibrated in the experimental conditions by an applied $J_I$ of 1.5 μA cm$^{-2}$. Recording procedure was as in the optrode measurements, with minor changes. Currents were acquired until a plateau peak was reached in the various ROI and times indicated, typically in <1 min. To mitigate the electromagnetic noise, we used a Faraday 'wall' (grounded aluminum-wrapped cardboard) covering the microscope. Data were acquired and extracted using WinWCP V4 (Strathclyde Electrophysiology Software) and treated using Excel.

**Statistical analysis**. Blinding was not employed in data acquisition, treatment, or statistical analysis. Statistical inference tests were used as appropriate and are annotated in figure captions. Pre-tests were conducted for normality (Kolmogorov-Smirnov, Shapiro-Wilk and/or D'Agostino and Pearson tests) and equal variances (F test) assumptions. If assumptions were not verified, the large sampling mitigated normality influence and the Welch's correction addressed unequal variances. Data are presented as mean ± s.e.m., with sample size (n, biological replicates or blots (Western) and flows (cytometry) of tens of tadpoles) indicated in figures and text. Data are additionally presented as median ± min to max (with outliers) in the case of boxplots. At least two independent batches of tadpoles or oocytes were used per readout (rare exceptions solely in Supplementary Fig. 1d, 15a,b). Differences were considered significant when $p < 0.05$ and level of significances were as follow: NS non-significant; *$p < 0.05$; **$p < 0.01$; and ***$p < 0.001$. Data treatment and visualization were performed using Excel, except for flow cytometry data where were used FlowJo and R (The R Foundation for Statistical Computing). Statistical tests were performed using GraphPad Prism 5 (GraphPad Software).

## Data availability

The data (structured or treated, representative, and raw) that support the findings of this study are embedded in the paper and its Supplementary Information. Additionally, all

other relevant data (discrete and numerical) are available and unrestricted from the corresponding authors upon reasonable request.

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

# ARTICLE

58. Bonello, S. et al. Reactive oxygen species activate the HIF-1alpha promoter via a functional NFkappaB site. *Arterioscler. Thromb. Vasc. Biol.* **27**, 755–761 (2007).

59. Minet, E. et al. Hypoxia-induced activation of HIF-1: role of HIF-1α-Hsp90 interaction. *FEBS Lett.* **460**, 251–256 (1999).

60. Kubis, H.-P., Hanke, N., Scheibe, R. J. & Gros, G. Accumulation and nuclear import of HIF1 alpha during high and low oxygen concentration in skeletal muscle cells in primary culture. *Biochim. Biophys. Acta* **1745**, 187–195 (2005).

61. Li, W. et al. Extracellular heat shock protein-90α: linking hypoxia to skin cell motility and wound healing. *EMBO J.* **26**, 1221–1233 (2007).

62. Ceradini, D. J. et al. Progenitor cell trafficking is regulated by hypoxic gradients through HIF-1 induction of SDF-1. *Nat. Med.* **10**, 858–864 (2004).

63. McCaig, C. D., Rajnicek, A. M., Song, B. & Zhao, M. Controlling cell behavior electrically: current views and future potential. *Physiol. Rev.* **85**, 943–978 (2005).

64. Guo, X. et al. The galvanotactic migration of keratinocytes is enhanced by hypoxic preconditioning. *Sci. Rep.* **5**, 1–13 (2015).

65. Shen, Y. et al. Diabetic cornea wounds produce significantly weaker electric signals that may contribute to impaired healing. *Sci. Rep.* **6**, 1–11 (2016).

66. Lakowicz, J. R. *Principles of Fluorescence Spectroscopy*, 277–330 (Springer, US, 2006).

67. Chatni, M. R., Li, G. & Porterfield, D. M. Frequency-domain fluorescence lifetime optrode system design and instrumentation without a concurrent reference light-emitting diode. *Appl. Opt.* **48**, 5528–5536 (2009).

68. Mishina, N. M. et al. in *Methods in Enzymology* (eds. Cadenas, E. & Packer, L.) **526**, 45–59 (Elsevier Inc., Amsterdam, 2013).

69. Pearl, E. J., Grainger, R. M., Guille, M. & Horb, M. E. Development of *Xenopus* resource centers: the national *Xenopus* resource and the european *Xenopus* resource center. *Genesis* **50**, 155–163 (2012).

70. Tsujioka, H., Kunieda, T., Katou, Y., Shirahige, K. & Kubo, T. Unique gene expression profile of the proliferating *Xenopus* tadpole tail blastema cells deciphered by rna-sequencing analysis. *PLoS ONE* **10**, 1–15 (2015).

## Acknowledgements

This work was supported by a NIH EY019101, an AFOSR (FA9550-16-1-0052) and, in part, by an Unrestricted Grant from Research to Prevent Blindness, Inc., University of California (UC), Davis, Ophthalmology. F.F. was supported by Fundação para a Ciência e Tecnologia (FCT) grant SFRH/BD/87256/2012 (majority of work) and by a staff research associate position in the laboratory (minority of work). We thank Dr. Andreia Gomes (Departamento de Biologia, CBMA, Universidade do Minho, Portugal) for support. We are grateful to Dr. Andrew L. Miller (Department of Biology, Hong Kong University of Science and Technology, China), Dr. Marko Horb (Director of the NXR, MBL) and Dr. Esther Pearl (NXR, MBL), and Mr. Alan M. Shipley (Applicable Electronics, LLC.) for generously hosting F.F.'s visit to MBL (Summer season 2014), for help in providing access to local research facilities and materials (SMOT and tadpoles (NXR RRID: SCR_013731) included), and for helpful comments. We are also grateful to Mr. Christopher A. Shipley, owner of Applicable Electronics, LLC., for having the courtesy to loan a SMOT system to the Zhao lab, and special thanks to Mr. Eric Karplus (Science Wares, Inc.), designer and programmer of the SMOT, for providing software, technical assistance and support for its efficient installation and operation. We are further grateful to Mr. Eric Karplus for helping with the mathematics and interpretation behind the equations used to calculate O$_2$ flux. We are thankful to Dr. Christopher Murphy (Department of Ophthalmology and Vision Sciences, UC Davis (R01 EY016134 and P30 EY12576)) and Dr. Paul Russell (Department of Surgical and Radiological Sciences, UC Davis) lab for kindly providing 17-DMAG and western blotting reagents and equipment for the experiments. We are also thankful to Iman Jalilian (Department of Surgical and Radiological Sciences, UC Davis) and Elias Barriga (Department of Cell and Developmental Biology, University College London, UK) for helpful comments about the western blotting technique. We thanks to Dr. Yunyun Shen (Department of Dermatology, UC Davis; Department of Occupational and Environmental Health, Zhejiang University, China) for kindly dispensing the euthanized mice and Dr. Li Ma (Department of Dermatology, UC Davis; Skin and Cosmetic Research Department, Shanghai Skin Disease Hospital, China) for helpful comments on the immunohistochemistry technique. We are grateful to Zhao lab members for helpful discussion.

## Author contributions

F.F., VK.R., G.L. and M.Z. designed the experiments. F.F. performed most experiments and analyzed the data and results. F.F., VK.R. and K.Z. performed and analyzed the western blots. F.F. and G.L. performed the flow cytometry; G.L. analyzed the flow cytometry. VK.R. and G.L. helped to interpret overall results. F.F., VK.R., G.L. and M.Z. outlined the manuscript. F.F. wrote the manuscript. All authors edited the manuscript.

## Additional information

**Competing interests:** The authors F.F. and M.Z. declare the following competing interests. The optrode measurements were acquired using turn-key systems provided free for use at MBL and loaned to the Zhao lab at no cost, by Applicable Electronics, LLC and Science Wares, Inc. The companies had no influence over the research, in design, execution, interpretation, or its reporting. No restrictions on data sharing have been imposed. The remaining authors declare no competing interests.

