## [Peer Review File · Nature Communications]

Reviewers' Comments:

Reviewer #1:

Remarks to the Author:

Ferriera et al investigated oxygen flux and ROS production during regeneration. The thesis of the work is that O₂ and ROS production collectively determine the partial pressure of O₂, and this impinges on regeneration. Presumably, injury affects both O₂ and ROS to generate a hypoxic state, and this somehow stabilizes HIF-1 α , which in turn activates transcription of regeneration permissive genes (e.g. HSP90). I think the MS addresses questions of interest to those working on tissue regeneration.

The profiling of O₂ during regeneration is one of the strengths of the study. The data show a correlation between O₂ influx and injury across several models, including the refractory stages of *Xenopus* regeneration, and this seems to also be associated with ROS availability as determined using an inhibitor of ROS production. Another strength of the study is the use of pharmaceutical inhibitors to define temporal windows within which Hif1- α and HSP90 appear to function, although it is not clear that these inhibitors adequately test an pO₂-dependent and ROS-independent mechanism of Hif1- α stabilization.

The paper does a good job of developing a model that links potential mechanisms together, but the data do not convincingly demonstrate these links to the point of excluding alternative interpretations. The correlation between O₂ and ROS production could have been investigated more deeply by directly measuring ROS. Without data for ROS, it is difficult to accept the premise that it is the balance of O₂ and ROS that fine tunes pO₂ in the regenerate and that this influences HIF1 α stability (directly or indirectly) and downstream gene expression. Is echinomycin treatment a good probe for pO₂ imbalance? Also, while HSP90 may act downstream of HIF1 α , another possibility is that HSP90 may be one of the factors that functions in the stabilization or destabilization of HIF1 α , placing it in parallel or upstream of HIF1 α . Further, HIF1 α and HSP90 inhibitor dosage, something that was not investigated, may differentially affect HIF1 α activity and perhaps in an pO₂-independent manner. How might HSP90 act to affect biological processes associated with regeneration? While the proposed simple model does fit data that were generated, the results are not definitive and can be interpreted differently. In this regard, the discussion is a bit too speculative and it seemed a bit much to integrate bioelectrical signaling into the final model when no studies were performed. The MS comes across as part part experimental and part review.

The statistical analyses were appropriate. It would be better to include drug concentrations in the body of the manuscript and figure legends.

Reviewer #2:

Remarks to the Author:

Comments on MS NCOMMS-17-07125

The authors report about the use of a commercially available Presens fiber optical chemo sensor (so called Optode) for monitoring the oxygen consumption in vertebrate regeneration assays. In principle optodes are ideal systems for the monitoring of low oxygen levels in aqueous systems and could be used as described in section "Optode Measurements" on page 17 for the vertebrate regeneration assays.

However, there are some major drawbacks from an analytical point of view:

i. The authors often use the term "mmHg" as a measure of pressure (e.g. in section "Results" as "160 mmHg" or "0.1 - 1% or 1 - 10 mgHg"). This should not be accepted. Pressure data should be given according to the IUPAC guidelines.

ii. The authors sometimes use "O₂ concentration", sometimes they use "pO₂", sometimes only "O₂". There are large differences between the three terms. Optodes can be used to monitor the dissolved oxygen concentration (often called DO). From a detailed point of view the oxygen has to permeate through a gas permeable membrane of the sensor patch out of the aqueous system to get in contact with the oxygen sensitive dye. Thus, the partial pressure is measured. I suggest to use only "pO₂".

iii. Normally optodes are calibrated via a "two point procedure (0 % pO₂: oxygen free aeration: 100% pO₂: aeration with defined oxygen concentrations e.g. air). As far as I understand the text, the authors calibrated with sulfite (0% pO₂) and air. For the sulfite procedure they didn't mention that cobalt ions (or similar) are necessary to establish an oxygen free medium. For the aeration process they mention a "20.95% O₂ solution". Normally this point is described at 100% pO₂ (maximum saturation). Here the paper is inaccurate. The amount of dissolved oxygen is drastically dependent on the composition of the aqueous phase. The Henry coefficient is describing this as a function of the composition of the medium. The dissolved oxygen concentration might vary tremendously. Was the Henry coefficient really constant during all experiments (ion strength; pH etc.)? If always the same medium composition is used in the experiments the procedure can be used as described.

In equation 2 the dimension on the left side is not equal to the dimension on the right side. Additionally on the left side [μM] is used and on the right side mol l⁻¹. Strange! On the right side you still have "%".

The authors should first describe the calibration procedures in detail and the assay system (medium composition etc.) in detail and in the second part of this chapter the calculations for the transfer of the optode data to the oxygen concentration should be presented before the flux analysis is given.

Conclusion:

In principle the optode system can be used as analytical tool for the experiments described. However, the authors are inaccurate and not consequent in describing the real procedure and especially the right terminology for dissolved oxygen concentrations. The defined terminology should be given and be used throughout the paper. The chapter "Optode Measurements" must be corrected and rewritten.

Reviewer #3:

Remarks to the Author:

This is an interesting manuscript, which aims to connect oxygen consumption, reactive oxygen species production, hypoxia and HIF1α activity during appendage regeneration in *Xenopus* tadpoles. The findings are interesting. However, there are several aspects of the work, which suggests that the relationship between oxygen influx, ROS, hypoxia and HIF1α activity may not be quite as straight forward as the paper would hope to suggest. In particular, the authors describe contradictory findings, which affect the conclusions and narrative they make. Thus, in my opinion further work needs to be done to solidify the conclusions that the authors wish to make.

Major comments / concerns:

1. The authors show that tail regeneration is associated with an increased oxygen influx into the regenerating tail. They go on to suggest that this oxygen influx fuels ROS production, as inhibiting the NADPH oxidases (and thus ROS production) decreases oxygen influx. However, they also show that, during the refractory stages (i.e. the non-regenerative stages) oxygen influx increases. However, the authors suggest that these stages are associated with lower ROS production (although they do not actually show this). Thus the question arises, why is oxygen influx higher

during the non-regenerative stages, if ROS production is expected to be lower at these stages? At this point, it becomes clear that the authors should assess ROS levels in the regenerative versus non-regenerative stages, rather than just assert that differences exist. Even if ROS production is lower in the non-regenerative stages, why is there higher oxygen influx at these stages? Higher oxygen influx would be expected to be associated with higher oxygen consumption. If so, what is consuming the oxygen? If not the NADPH oxidases, is it higher OX/PHOS in the mitochondria, which is responsible? Minimally the authors should assess ROS levels in the regenerative versus non-regenerative phases, rather than simply stating (without evidence) that the non-regenerative phases have lower ROS levels (despite the higher oxygen influx). There are several methods for assaying ROS levels, which include the use of redox sensitive dyes and also transgenic lines, such as HyperYFP.

2. Secondly, while the authors perform experiments addressing the role of HIF1alpha during tail regeneration, the authors do not assess whether HIF1alpha is controlled by hypoxia during regeneration (as opposed to other mechanisms independent of hypoxia). Yet, this is the conclusion they wish to make. The evidence the authors present in the manuscript is all based on inhibitors and activators of HIF1alpha, which may or may not reflect regulation by hypoxia. Indeed the authors do not look at HIF1alpha stabilization in their experiments, nor do they look at HIF1alpha expression (both of which could be regulated by ROS in a hypoxia independent manner). The fact that regeneration cannot be rescued in the non-regenerative stages by hypoxia would seem to suggest that HIF1alpha activity during regeneration is not regulated by hypoxia. Also, the fact that DMOG cannot rescue regeneration in the non-regenerative stages would seem to suggest that HIF1alpha activation is not sufficient for regeneration (and thus the authors cannot claim that HIF1alpha activity is sufficient for regeneration, as suggested by the title of the section at the bottom of page 8 of the manuscript).

In summary, there are too many loose ends in the findings for the conclusions the authors have tried to make in the manuscript. Minimally, the authors should assess the following, and based on the result, they should be able to make a more compelling case as what conclusion they wish to draw from their work:

- Assay ROS levels following the various perturbations in the regenerative and non-regenerative stages
- Assay HIF1alpha protein levels (stability) following the various perturbations in the regenerative and non-regenerative stages
- Assay HIF1alpha expression levels following the various perturbations in the regenerative and non-regenerative stages
- Determine whether the tissues are hypoxic (or normoxic) following the various perturbations in the regenerative and non-regenerative stages

Response to Referees Letter

“Early redox activities modulate vertebrate regeneration” by Fernando Ferreira *et al.*

Unrequested edit

Bar plots were converted into boxplots to highlight data distribution, which maximizes the information that can be extracted from data visualization. Note that data was unaltered in the process. Converted plots are Fig. 2b, 3b,c,e. In the Supplementary information, most bar plots were also converted into boxplots; exceptions were decided based on representing key information with clarity.

Response to Reviewer #1

Reviewer #1 (Remarks to the Author)

Ferreira et al investigated oxygen flux and ROS production during regeneration. The thesis of the work is that O₂ and ROS production collectively determine the partial pressure of O₂, and this impinges on regeneration. Presumably, injury affects both O₂ and ROS to generate a hypoxic state, and this somehow stabilizes HIF-1 α , which in turn activates transcription of regeneration permissive genes (e.g. HSP90). I think the MS addresses questions of interest to those working on tissue regeneration.

The profiling of O₂ during regeneration is one of the strengths of the study. The data show a correlation between O₂ influx and injury across several models, including the refractory stages of *Xenopus* regeneration, and this seems to also be associated with ROS availability as determined using an inhibitor of ROS production. Another strength of the study is the use of pharmaceutical inhibitors to define temporal windows within which Hif1- α and HSP90 appear to function, although it is not clear that these inhibitors adequately test an pO₂-dependent and ROS-independent mechanism of Hif1- α stabilization.

The paper does a good job of developing a model that links potential mechanisms together, but the data do not convincingly demonstrate these links to the point of excluding alternative interpretations. The correlation between O₂ and ROS production could have been investigated more deeply by directly measuring ROS. Without data for ROS, it is difficult to accept the premise that it is the balance of O₂ and ROS that fine tunes pO₂ in the regenerate and that this influences HIF1 α stability (directly or indirectly) and downstream gene expression.

A.: We thank the reviewer for the positive and insightful comments. We have done new experiments to measure ROS using transgenic animals (also in response to Reviewer #3's comment). These data show significant differences (unpublished until now) between ROS levels in regenerative vs. refractory period tadpoles that robustly correlate with regeneration (see new Fig. S7 and Results Lines 164-170).

In the other conditions of the present study, we have performed experiments to establish the relationship between regeneration and ROS production using the same conditions (species, age, timings, regions of interest and drugs) in an earlier study (Ferreira *et al.*, *Development* **143**, 4582–4594 (2016)). It is important to note that our experiments are in concurrence with previously published (by another laboratory) comprehensive data on spatiotemporal profiling of ROS (specifically H₂O₂) in *Xenopus* spp. using very similar experimental conditions (Love *et al.*, *Nat. Cell Biol.* **15**, 222–8 (2013)). The new and previously published data show a robust correlation of ROS with regeneration that are effectively integrated in the

revised manuscript (see new Fig. S7 and Results Lines 154, 155; 164-172; 180, 181; highlighted in red colored font).

Importantly, we now provide quantification and imaging data for hypoxia and HIF-1 α stability (protein) levels (see new Fig. 5 and new Results section in Line 234).

Is echinomycin treatment a good probe for pO₂ imbalance?

A.: Echinomycin blocks the binding of HIF-1 α , thereby limiting its transcriptional activity. Therefore, this drug inhibits available HIF-1 α regardless of pO₂ (im)balance. Naturally, if pO₂ is normoxic, fewer HIF-1 α will be available for the drug to act, because of O₂-dependent degradation. We used echinomycin not to disturb the presumptive hypoxia, but rather to prove that HIF-1 α transcriptional activity is required for regeneration, and for this finality results were very significant. We are unaware of drugs that are specific to inhibiting HIF-1 α and at the same time serve as a good probe of pO₂ (im)balance.

Also, while HSP90 may act downstream of HIF1 α , another possibility is that HSP90 may be one of the factors that functions in the stabilization or destabilization of HIF1 α , placing it in parallel or upstream of HIF1 α . Further, HIF1 α and HSP90 inhibitor dosage, something that was not investigated, may differentially affect HIF1 α activity and perhaps in an pO₂-independent manner.

A.: We agree with the reviewer that HSP90 may act in parallel or upstream of HIF-1 α and considered this. In order to establish whether HSP90 was in fact downstream of HIF-1 α we used terazosin (an α 1-adrenergic receptor agonist) to rescue echinomycin-impaired regeneration. However, terazosin is not a specific activator of HSP90 (although it has been used to activate HSP90 to promote stress resistance; Chen *et al.*, *Nat Chem Biol*, **11**(1), 19-25 (2015)) and, as such, the results were inconclusive to reliably establish causality. An extensive search to identify a targeted activator of HSP90 resulted in little results. To the best of our knowledge, currently there are no specific small molecule activators of HSP90 available commercially. Therefore, performing an experiment to establish the relationship between HIF-1 α and HSP90 without genetic manipulations is challenging. In the original and revised submission, we did not dismiss other epistasis possibilities; from our data and in congruence with the wounding healing literature, we propose that intracellular HSP90 putatively acts alongside HIF-1 α , and that its extracellularly secreted form (eHSP90 α) is a downstream target of HIF-1 α . Future studies will be required to establish this causality. These discussion points are clarified in the revised manuscript (Lines 444-463; highlighted in red colored font).

Further, for all drugs used, extensive dose-exposure testing was performed, although we did not always show in the original submission for clarity and simplicity. These have now been included within supplementary information (see new Fig. S8, S10-12 and S17). As noted in these figures, selected doses were based on the maximal penetrance in regeneration with minimal observable toxicity and developmental side defects, *i.e.*, selections were phenotype-based and not based in discrete effects on molecular targets.

How might HSP90 act to affect biological processes associated with regeneration?

A.: Considerable details of this are in the Discussion (Lines 444-463; highlighted in red colored font). Particularly, we emphasize the role of HSP90 in cell migration, as cell migration is essential for regeneration.

While the proposed simple model does fit data that were generated, the results are not definitive and can be interpreted differently. In this regard, the discussion is a bit too speculative and it seemed a bit much to integrate bioelectrical signaling into the final model when no studies were performed. The MS comes across as part part experimental and part review.

A.: We performed an extensive study comprehensively integrating bioelectric and redox state during regeneration (Ferreira *et al.*, *Development* **143**, 4582–4594 (2016)). In that study we robustly proved that ROS modulate various bioelectric activities during regeneration of the same model. The bioelectric analysis in the submitted manuscript follows the footprints of the previous study and intends to be a starting point for an even larger integration of redox-bioelectric activities during regeneration, aiming to merge together ROS, HIFs, O₂ and the bioelectric parameters membrane and transepithelial potential, electric currents and electric fields. Therefore, we think that this is a significant piece of data with statistical power, setting the stage for future work.

To highlight this and make the manuscript fully experimental following surgical reviewer's comment, we moved the key result from Supplementary information to new Fig. 7b, wrote a new Results section (Line 311; highlighted in red colored font) and revised discussion accordingly (Lines 473-482; highlighted in red colored font).

The statistical analyses were appropriate. It would be better to include drug concentrations in the body of the manuscript and figure legends.

A.: We respect the reviewer's suggestion to include drug concentrations in the body, legends and figures. However, to avoid confusion and to maintain simplicity, we choose to annotate concentrations and timings in plots only but not in legends. Further, results described in the main text where mostly when drugs were used at a singular concentration.

That being said, complying with the request, where essential, we have now added drug/media concentrations to figure legends (cases of Fig. 1, 2 and 5) for ease of reading.

Response to Reviewer #2

Reviewer #2 Remarks to the Author

Comments on MS NCOMMS-17-07125

The authors report about the use of a commercially available Presens fiber optical chemo sensor (so called Optode) for monitoring the oxygen consumption in vertebrate regeneration assays. In principle optodes are ideal systems for the monitoring of low oxygen levels in aqueous systems and could be used as described in section “Optode Measurements” on page 17 for the vertebrate regeneration assays.

However, there are some major drawbacks from an analytical point of view:

i. The authors often use the term “mmHg” as a measure of pressure (e.g. in section “Results” as “160 mmHg” or “0.1 - 1% or 1 – 10 mgHg”). This should not be accepted. Pressure data should be given according to the IUPAC guidelines.

A.: We thank the reviewer for the positive and insightful comments. As recommended, we have converted all pressure values to Pascal units throughout the manuscript (highlighted in red colored font). As much of the literature known to us and cited in this study use mmHg and %, we also noted those to facilitate reading and expedite understanding to a broader community.

ii. The authors sometimes use “O₂ concentration”, sometimes they use “pO₂”, sometimes only “O₂”. There are large differences between the three terms. Optodes can be used to monitor the dissolved oxygen concentration (often called DO). From a detailed point of view the oxygen has to permeate through a gas permeable membrane of the sensor patch out of the aqueous system to get in contact with the oxygen sensitive dye. Thus, the partial pressure is measured. I suggest to use only “pO₂”.

A.: We understand the concern of the reviewer and revised the manuscript and corrected all the possibly misleading terminology. The use of ‘O₂ concentration’ term appears in Materials and Methods to detail an equation (Line 561); it does not appear in the remainder of the manuscript. In the original and revised manuscript, the use of ‘pO₂’ term refers to static measurements (in this manuscript only mentioned from literature) and raw data (before conversion to flux; only mentioned in supplementary information – Fig. S5, S6), whereas ‘O₂’ term refers more to flux measurements (after conversion to flux). We are well aware that we always measured dissolved oxygen (although it is possible to measure O₂ in the gaseous phase with optrode (data not shown)); to make this explicit we added the text ‘dissolved O₂’ to the Materials and Methods (Line 533). We think that it is not appropriate to use by default the term ‘pO₂’, mainly because we present the data as flux units.

iii. Normally optodes are calibrated via a “two point procedure (0 % pO₂: oxygen free aeration: 100% pO₂: aeration with defined oxygen concentrations e.g. air). As far as I understand the text, the authors calibrated with sulfite (0% pO₂) and air. For the sulfite procedure they didn’t mention that cobalt ions (or similar) are necessary to establish an oxygen free medium. For the aeration process they mention a “20.95% O₂ solution”. Normally this point is described at 100% pO₂ (maximum saturation). Here the paper is inaccurate. The amount of dissolved oxygen is drastically dependent on the composition of the

aqueous phase. The Henry coefficient is describing this as a function of the composition of the medium. The dissolved oxygen concentration might vary tremendously. Was the Henry coefficient really constant during all experiments (ion strength; pH etc.)? If always the same medium composition is used in the experiments the procedure can be used as described.

A.: As the reviewer correctly noted we used a two-point calibration of free and saturated pO_2 . For the free pO_2 , we used a sodium bisulfite solution (mixture of $NaHSO_3$ and $Na_2S_2O_5$). When establishing the technique in the lab, we also used nitrogen-purged water (aeration with $N_2(g)$ 100%); multiple calibrations with N_2 gas gave the same output as the sulfite solution (see plot below). Thus, we calibrated confidently with the sulfite solution. Saturated DO was achieved using air aeration of deionized water. As recommended, we changed terminology to 0% pO_2 and 20.95% pO_2 (Line 542, 543). Daily calibrations were consistent over this study and also for other ongoing studies.

The reviewer is correct to identify that salinity can affect DO. In this study the ionic concentrations and pH were always the same across experiments, and as such no compensation to correct for salinity changes was deemed essential.

In equation 2 the dimension on the left side is not equal to the dimension on the right side. Additionally on the left side $[\mu M]$ is used and on the right side mol l-1. Strange! On the right side you still have “%”.

A.: We have deeply and carefully proof-read all used equations and units and read and analyzed the complementary literature for the same equations (namely the widely used textbook *Principles of Fluorescence Spectroscopy* 3rd edition by Lakowicz, 2006), and the mathematics is, to our best knowledge, correct. The written equation 1 (Line 563) is just a minor adaptation of a previously published equation and data analysis (Chatni *et al. Appl. Opt.* **48**, 5528–36 (2009)). Also, this equation is present in the instructions manuals of optrode manufactures in their sections on how to derive O_2 concentrations from optrode’s raw data (*e.g.*, see PreSens’ instruction manual of Fibox 3-Trace – print screen below where our equation is based on equation #18 therein). In addition to the literature, we consulted actively these manuals in order to fully understand the equations and respective components. The equation 1’s component “ V_M ” is molar volume and its units is “ $l\ mol^{-1}$.” In the manuscript, the equation 1’s component $(pO_2/20.2095)/100$ is the ratio of O_2 in the gas mixture referred in the publication above as Q . The “ pO_2 ” factor is the extracted raw data from the ASET interface software, derived as follows:

The ASET software communicates with electronic firmware that has an embedded microprocessor that automatically and in real-time calculates pO_2 (%) using the fluorescence lifetime-based method following

the Stern-Volmer equation based on a measurement of phase angle shift relative to calibration values obtained at known temperature and pressure for the sensor being used. This information and supporting references are now added in detail to the revised manuscript (see Lines 557-561).

Oxygen Concentration

in mg/L

$$c_{O_2} [\text{mg/L}] = \frac{P_{\text{atm}} - P_{\text{W}}(T)}{P_{\text{N}}} \cdot \frac{\% \text{ air - saturation}}{100} \cdot 0.2095 \cdot \alpha(T) \cdot 1000 \cdot \frac{M(O_2)}{V_M} \quad (16)$$

in ppm

$$c_{O_2} [\text{ppm}] = \frac{P_{\text{atm}} - P_{\text{W}}(T)}{P_{\text{N}}} \cdot \frac{\% \text{ air - saturation}}{100} \cdot 0.2095 \cdot \alpha(T) \cdot 1000 \cdot \frac{M(O_2)}{V_M} \quad (17)$$

in $\mu\text{mol/L}$

$$c_{O_2} [\mu\text{mol/L}] = \left[\frac{P_{\text{atm}} - P_{\text{W}}(T)}{P_{\text{N}}} \cdot \frac{\% \text{ air - saturation}}{100} \cdot 0.2095 \cdot \alpha(T) \cdot 1000 \cdot \frac{M(O_2)}{V_M} \right] \cdot 31.25 \quad (18)$$

P_{atm} : actual atmospheric pressure;
 P_{N} : standard pressure (1013 mbar);
 0.2095: volume content of oxygen in air;
 $P_{\text{W}}(T)$: vapor pressure of water at temperature T given in Kelvin;
 $\alpha(T)$: Bunsen absorption coefficient at temperature T;
 $M(O_2)$: molecular mass of oxygen (32 g/mol);
 V_M : molar volume (22.414 l/mol);

Moreover, we used an alternative equation to validate and to be confident that the mathematics was indeed correct. The alternative equation was based on the DO concentration for 100% air saturated water at sea level (1013.25 mbar) and 23 °C (room temperature). DO for these conditions is $\sim 8.53 \text{ mg l}^{-1}$ ($= 266517.81 \text{ pmol cm}^{-3}$). With the assumption that 100% air saturation equals to 20.95% pO_2 , then we can calculate any observed O_2 concentration in mg l^{-1} or pmol cm^{-3} as $[O_2] (\text{pmol cm}^{-3}) = (266517.81 \cdot pO_2) / 20.95$; pO_2 is the extracted raw data (%) from the firmware equations of the ASET interface software. Then, having both near and far concentrations values, we used them in the Fick's first law of diffusion and obtained the flux values. This alternative equation gave the same flux values as the equation provided in the manuscript (see plot below).

Altogether, we were and are very confident in the correctness of the equation provided in the manuscript, which follows literature and is recommended by optrode manufactures. To further validate equation and the technique itself, we performed several artificial sinks using N₂ gas bubbles (see plot below). The sinks provided conservative and expected exponential influxes robustly validating equation and technique.

The calibration and artificial sink data shown above are among the data being prepared for a technical manuscript specifically about the optrode measurements to be submitted soon and is out of the scope of the current manuscript.

The authors should first describe the calibration procedures in detail and the assay system (medium composition etc.) in detail and in the second part of this chapter the calculations for the transfer of the optode data to the oxygen concentration should be presented before the flux analysis is given.

A.: In addition to the responses above, the methodology and procedures are now described in detail (Lines 532-583; highlighted in red colored font).

Conclusion:

In principle the optode system can be used as analytical tool for the experiments described. However, the authors are inaccurate and not consequent in describing the real procedure and especially the right terminology for dissolved oxygen concentrations. The defined terminology should be given and be used throughout the paper. The chapter "Optode Measurements" must be corrected and rewritten.

A.: We edited the manuscript and the section "Optrode measurement" (Lines 532-583; highlighted in red colored font) to address reviewer's concerns.

Response to Reviewer #3

Reviewer #3 Remarks to the Author

This is an interesting manuscript, which aims to connect oxygen consumption, reactive oxygen species production, hypoxia and HIF1 α activity during appendage regeneration in *Xenopus* tadpoles. The findings are interesting. However, there are several aspects of the work, which suggests that the relationship between oxygen influx, ROS, hypoxia and HIF1 α activity may not be quite as straightforward as the paper would hope to suggest. In particular, the authors describe contradictory findings, which affect the conclusions and narrative they make. Thus, in my opinion further work needs to be done to solidify the conclusions that the authors wish to make.

Major comments / concerns:

1. The authors show that tail regeneration is associated with an increased oxygen influx into the regenerating tail. They go on to suggest that this oxygen influx fuels ROS production, as inhibiting the NADPH oxidases (and thus ROS production) decreases oxygen influx. However, they also show that, during the refractory stages (i.e. the non-regenerative stages) oxygen influx increases. However, the authors suggest that these stages are associated with lower ROS production (although they do not actually show this). Thus the question arises, why is oxygen influx higher during the non-regenerative stages, if ROS production is expected to be lower at these stages? At this point, it becomes clear that the authors should assess ROS levels in the regenerative versus non-regenerative stages, rather than just assert that differences exist. Even if ROS production is lower in the non-regenerative stages, why is there higher oxygen influx at these stages? Higher oxygen influx would be expected to be associated with higher oxygen consumption. If so, what is consuming the oxygen? If not the NADPH oxidases, is it higher OX/PHOS in the mitochondria, which is responsible? Minimally the authors should assess ROS levels in the regenerative versus non-regenerative phases, rather than simply stating (without evidence) that the non-regenerative phases have lower ROS levels (despite the higher oxygen influx). There are several methods for assaying ROS levels, which include the use of redox sensitive dyes and also transgenic lines, such as HyperYFP.

A.: We thank the reviewer for the positive and insightful comments. We agree with the reviewer's comments and have duly performed the requested experiments. The results are now added to supplementary information (see new Fig. S7 and Results Lines 164-172; highlighted in red colored font).

In response to the question of 'why' refractory period has a higher O₂ influx than the regenerative period, we have attempted to provide a comprehensive argument/hypothesis in the Discussion (Lines 397-409; some text highlighted in red colored font). Briefly, low ROS coupled with minimal/no proliferation and bud maturation, in the refractory period, suggest low O₂ demand. The animal growth (influx magnitude for cutaneous respiration diminishes with size) and bud size were effectively discarded as reasons for the higher influxes (Fig. 1c, Fig. S3). Moreover, similar influx magnitudes in regenerative and refractory fin measurements point to bud-specific and thus regeneration-specific O₂ fluxes. The elusive explanation is thus probably related with the regeneration process itself. Mitochondria activity – as well pointed by the reviewer – might be a prime candidate and its role requires further investigation.

It is important to note that the higher influx in the refractory period provide an explanation for the abrogated regeneration, as the large O₂ influx and lower O₂ consumption disrupts the local hypoxic

microenvironment preventing the required HIF-1 α activity. New data presented in the revised manuscript pertinent to hypoxia and HIF-1 α measurements support this (see new Fig. 5; also new Fig. S19). We discussed this point in the same Discussion paragraph, now revised (Lines 397-409; some text highlighted in red colored font).

2. Secondly, while the authors perform experiments addressing the role of HIF1alpha during tail regeneration, the authors do not assess whether HIF1alpha is controlled by hypoxia during regeneration (as opposed to other mechanisms independent of hypoxia). Yet, this is the conclusion they wish to make. The evidence the authors present in the manuscript is all based on inhibitors and activators of HIF1alpha, which may or may not reflect regulation by hypoxia. Indeed the authors do not look at HIF1alpha stabilization in their experiments, nor do they look at HIF1alpha expression (both of which could be regulated by ROS in a hypoxia independent manner). The fact that regeneration cannot be rescued in the non-regenerative stages by hypoxia would seem to suggest that HIF1alpha activity during regeneration is not regulated by hypoxia. Also, the fact that DMOG cannot rescue regeneration in the non-regenerative stages would seem to suggest that HIF1alpha activation is not sufficient for regeneration (and thus the authors cannot claim that HIF1alpha activity is sufficient for regeneration, as suggested by the title of the section at the bottom of page 8 of the manuscript).

A.: We understand the reviewer's concern here and we now provide new data detailing hypoxia and HIF-1 α stability levels in the different conditions (see new Fig. 5). The new data show that HIF-1 α is most likely stabilized by hypoxia and not by ROS (see new Results section in Line 234 and revised Discussion paragraph in Lines 419-443; highlighted in red colored font).

Our original submission data demonstrates that HIF- 1 α stabilization using the hypoxia-mimicking drug DMOG did, in fact, induce regeneration in the non-regenerative refractory period tadpoles (Fig. 4b,b').

In addition to the strong correlation between hypoxia and HIF-1 α , another empirical evidence points to hypoxia as the cause of HIF-1 α stabilization. Epistasis assays from the original submission (Fig. 6) demonstrate that ROS does not directly activate HIF-1 α to modulate regeneration. However, and as we discuss lengthily in the revised manuscript, this does not exclude an indirect stabilization of HIF-1 α , because of the O₂ consumption to produce ROS. Newly added data (see new Fig. 5 and new Fig. S15) show that inhibited ROS production decreases hypoxia and protein levels in a proportional penetrance; further ROS scavenging (which does not affect O₂ influx) neither impair hypoxia nor HIF-1 α levels. These together imply that the effect of ROS on HIF-1 α protein levels is due to hypoxia *via* the balance of O₂ consumption and O₂ influx, demonstrating that HIF-1 α is stabilized by local hypoxia. These are now described in detail in the revised Discussion (Lines 419-443; highlighted in red colored font).

In summary, there are too many loose ends in the findings for the conclusions the authors have tried to make in the manuscript. Minimally, the authors should assess the following, and based on the result, they should be able to make a more compelling case as what conclusion they wish to draw from their work:

- Assay ROS levels following the various perturbations in the regenerative and non-regenerative stages
A.: As suggested, we have now included data revealing significant differences in ROS in regenerative *vs.* refractory period tadpoles in the revised manuscript (see new Fig. S7 and Results Lines 164-172;

highlighted in red colored font), revealing significant differences unpublished before. As of DPI effect on ROS, we (Ferreira *et al.*, *Development* **143**, 4582–4594 (2016)) and others (Love *et al.*, *Nat. Cell Biol.* **15**, 222–8 (2013)) published comprehensive measurements before in the same species and conditions.

- Assay HIF1alpha protein levels (stability) following the various perturbations in the regenerative and non-regenerative stages

A.: This data is added to the revised manuscript (see new Fig. 5c-d; new Results section in Line 234; and Discussion Lines 419-443; highlighted in red colored font). We selected 1 hpa, as it is the time-window when the influence of HIF-1 α on regeneration was greatest.

- Assay HIF1alpha expression levels following the various perturbations in the regenerative and non-regenerative stages

A.: While we generally agree that expression levels may differ from protein levels, it is important to note that, in the context of regeneration we are mainly interested in the post-translational function of HIF-1 α . Post-translational modification of HIF-1 α can be stabilized (by hypoxia in our case) to dimerize with HIF-1 β in the nucleus to behave as a transcription factor. As such we do not include mRNA analysis in this study, although it may be more relevant in a future study.

- Determine whether the tissues are hypoxic (or normoxic) following the various perturbations in the regenerative and non-regenerative stages

A.: This data is added to the revised manuscript (see new Fig. 5a-b; new Results section in Line 234; and Discussion Lines 419-443; highlighted in red colored font). We selected 1 hpa, as it is the time-window when the influence of HIF-1 α on regeneration was greatest.

Reviewers' Comments:

Reviewer #1:

Remarks to the Author:

The revised manuscript adequately addressed my comments/concerns. In addressing early post-injury events, the results provide interesting new insights about relationships among O₂, ROS, hypoxia, and bioelectricity that will likely influence thinking in the field. I do suggest that the authors carefully re-read the new passages in the revised manuscript as there are a significant number of grammatical mistakes that make it difficult to follow lines of reasoning.

Reviewer #2:

Remarks to the Author:

I carefully went through the revised manuscript and would like to thank the authors for the excellent revision effort to the comments especially in regard of "mmHg" and "pO₂".

I can understand that in medicine "mmHg" is still used, but for me as an analytical chemist we should follow the IUPAC rules. The paper is now corrected regarding these comments and I have the feeling that researchers from Natural Sciences as well as from Medicine can now easily follow this valuable paper.

Regarding my comments on the different equations I would like to thank the authors for their explanations. The equations given by them are correct when you follow the cited papers. The equation is in principle usable. Unfortunately, there is a mistake in the Fibox 3-Trace instruction manual which they added to their comments.. I already contacted PreSens to correct this. The content of the paper stands for its own and can be regarded as correct.

In conclusion I can state that all my comments were regarded in the revised versions. All the new input is fully convincing. I thus comment the publication of this paper in Nature Communications.

Reviewer #3:

Remarks to the Author:

The authors have performed a number of experiments aimed at addressing the reviewers' comments. As a result, the manuscript has been much improved.

One set of experiments that both Reviewer 1 and 3 requested was a thorough assessment of ROS levels in regenerative and non-regenerative stages. The authors present new experiments aimed at comparing ROS levels in regenerative and non-regenerative stages, using a transgenic line provided by a stock center (data presented in new supplementary figure 7). However, it is unclear how the ROS levels were performed and analysed in these experiments. The ROS genetic sensor used, HyPer, is a ratiometric genetic sensor, and thus, normally ROS levels using this probe are shown as ratios. The authors present the data as arbitrary units and essentially no fluorescence in the images is considered to reflect low ROS levels (see Fig S7b). In fact, HyPer is fluorescent both in the presence or absence of hydrogen peroxide, and thus, no fluorescence suggests no or low HyPer expression, rather than low ROS levels being present. The authors should provide more information about how these data were obtained and analysed. As it stands, the ROS imaging data presented are confusing and unsatisfactory, and given that these data were deemed critical for two of the reviewers, it is still difficult to recommend acceptance of the manuscript based on these new inconclusive data.

Minor comments:

1. The two new sentences in the abstract (highlighted in red in the revised manuscript) are confusing and awkward, as written. Consider revising.
2. The order of supplementary figures are not the same as when they are first referred to in the main text.

Response to Referees' Letter (II)

“Early redox activities modulate vertebrate regeneration” by Fernando Ferreira *et al.*

Response to Reviewer #1

Reviewer #1 (Remarks to the Author)

The revised manuscript adequately addressed my comments/concerns. In addressing early post-injury events, the results provide interesting new insights about relationships among O₂, ROS, hypoxia, and bioelectricity that will likely influence thinking in the field. I do suggest that the authors carefully re-read the new passages in the revised manuscript as there are a significant number of grammatical mistakes that make it difficult to follow lines of reasoning.

A.: We thank the reviewer for the positive comments and assessment. We have thoroughly revised the passages to correct for any grammatical mistakes (highlighted in red colored font).

Response to Reviewer #2

Reviewer #2 (Remarks to the Author)

I carefully went through the revised manuscript and would like to thank the authors for the excellent revision effort to the comments especially in regard of “mmHg” and “pO₂”. I can understand that in medicine “mmHg” is still used, but for me as an analytical chemist we should follow the IUPAC rules. The paper is now corrected regarding these comments and I have the feeling that researchers from Natural Sciences as well as from Medicine can now easily follow this valuable paper. Regarding my comments on the different equations I would like to thank the authors for their explanations. The equations given by them are correct when you follow the cited papers. The equation is in principle usable. Unfortunately, there is a mistake in the Fibox 3-Trace instruction manual which they added to their comments. I already contacted PreSens to correct this. The content of the paper stands for its own and can be regarded as correct. In conclusion I can state that all my comments were regarded in the revised versions. All the new input is fully convincing. I thus comment the publication of this paper in Nature Communications.

A.: We thank the reviewer for the encouraging comments and positive assessment.

Response to Reviewer #3

Reviewer #3 (Remarks to the Author)

The authors have performed a number of experiments aimed at addressing the reviewers' comments. As a result, the manuscript has been much improved. One set of experiments that both Reviewer 1 and 3 requested was a thorough assessment of ROS levels in regenerative and non-regenerative stages. The authors present new experiments aimed at comparing ROS levels in regenerative and non-regenerative stages, using a transgenic line provided by a stock center (data presented in new supplementary figure 7). However, it is unclear how the ROS levels were performed and analysed in these experiments. The ROS genetic sensor used, HyPer, is a ratiometric genetic sensor, and thus, normally ROS levels using this probe are shown as ratios. The authors present the data as arbitrary units and essentially no fluorescence in the

images is considered to reflect low ROS levels (see Fig S7b). In fact, HyPer is fluorescent both in the presence or absence of hydrogen peroxide, and thus, no fluorescence suggests no or low HyPer expression, rather than low ROS levels being present. The authors should provide more information about how these data were obtained and analysed. As it stands, the ROS imaging data presented are confusing and unsatisfactory, and given that these data were deemed critical for two of the reviewers, it is still difficult to recommend acceptance of the manuscript based on these new inconclusive data.

A.: We thank the reviewer for the insightful comments. We understand the reviewer's concerns and we now provide additional details in the revised manuscript (see updated caption of Supplementary Fig. 7a-b' on page 11, and expanded Supplementary Methods, subsection 'H₂O₂ fluorescence imaging' on page 27). Specific details for the points raised by the reviewer can be found in those places, highlighted in red colored font.

As the reviewer rightfully pointed out, HyPer is a genetically encoded ROS sensor that has two excitation peaks at 420 nm and 500 nm, but one emission peak at 516 nm in the absence of H₂O₂. In the presence of H₂O₂ the fluorescence at 420 nm maintains and proportionally increases at 500 nm, thus allowing for ratiometric measurements of H₂O₂.

Indeed, in our study, we semi-quantitatively determined fluorescence intensities in the tadpoles using excitations of 405 (Alexa Fluor 405 channel) and 488 (Alexa Fluor 488 channel) nm and detected fluorescence with emission maxima set to 515 nm for both excitation wavelengths. This is now included in the updated Supplementary Methods on page 27: *"The values obtained with ex/em at 405/515 nm were always very low (thus considered as background fluorescence) and did not shift across conditions (ROS/H₂O₂/amputation); although fluorescence intensities varied markedly across conditions in the 488/515 nm ex/em spectra. Thus, a ratio of intensities at [488/515 nm] to [405/515 nm] would have resulted in very large values that may falsely be perceived artifactual. Therefore, for clarity, we only present the values for fluorescence obtained with ex/em at 488/515 nm and considered the fluorescence values in the ex/em at 405/515 nm spectra as background intensities in this study."* These quantifications are represented in the figures.

In fact, using such methods for control experiments, we confirmed significantly elevated fluorescence in the 488/515 ex/em nm spectra with H₂O₂ stimulation and no fluorescence in the 488/515 nm spectra for the non-transgenesis tails (see Supplementary Fig. 7a). Acknowledging this technical limitation, and to further support our findings regarding the differential ROS levels in regenerative and refractory periods we used a ROS-sensitive dye to image and semi-quantify ROS production in both periods. For these secondary validation experiments for the generation of ROS we used a chloromethyl derivative of H₂DCFDA (a very widely used general ROS/oxidative stress indicator). This analysis supports our HyPer findings, making them more robust (see new Supplementary Fig. 7c,c' and added caption on page 11, and see new subsection 'ROS fluorescence imaging' in Supplementary Methods on pages 27 and 28; see also Lines 175-177; highlighted in red colored font).

Minor comments:

1. The two new sentences in the abstract (highlighted in red in the revised manuscript) are confusing and awkward, as written. Consider revising.

A.: We have now revised these sentences in the abstract (see Lines 7-10; highlighted in red colored font).

2. The order of supplementary figures are not the same as when they are first referred to in the main text.

A.: We have strived to cite Supplementary Figures in order for most of the manuscript; however, occasionally, some order was deferred in favor of a better organized and structured Supplementary Information. We think that the compromise actually aids the reader to better understand the narrative in the manuscript. A key factor for the non-sequential citation of Supplementary Figures is the localization of Methods section in the end of the manuscript, after Results and Discussion (following journal guidelines). Then, naturally, for example, a Supplementary Figure or Panel simultaneously related with methodology and with an early figure will appear early on Results section and then late in the Methods section.

Reviewers' Comments:

Reviewer #2:

Remarks to the Author:

I was asked to take a look at the revised manuscript under the special aspects of reviewer #3

- The explanation and information given by the authors to the "HyPer-experiments" (HyPer is a genetically encoded ROS sensor) is from my point of view convincing. I can follow their information. The data given in the figures are helpful. The explanation of the analytical procedure is profound.
- I went through the figures of the supplementary section. I cannot see any problems even when the correlation to the main text is sometimes not very obvious. However, an extra revision seems not to be necessary.
- But (!!) the new sentences in the abstract (line 7 – 10) are still not clear. The structure is still confusing to me. Please carefully go through the whole abstract and polish it. The two sentences and the information given should be better integrated. Here they are written in a complete different way.

Final Response to Referee' Letter

“Early redox activities modulate vertebrate regeneration” by Fernando Ferreira *et al.*

Response to Reviewer #2

Reviewer #2 (Remarks to the Author)

I was asked to take a look at the revised manuscript under the special aspects of reviewer #3

- The explanation and information given by the authors to the “HyPer-experiments” (HyPer is a genetically encoded ROS sensor) is from my point of view convincing. I can follow their information. The data given in the figures are helpful. The explanation of the analytical procedure is profound.
- I went through the figures of the supplementary section. I cannot see any problems even when the correlation to the main text is sometimes not very obvious. However, an extra revision seems not to be necessary.
- But (!!) the new sentences in the abstract (line 7 – 10) are still not clear. The structure is still confusing to me. Please carefully go through the whole abstract and polish it. The two sentences and the information given should be better integrated. Here they are written in a complete different way.

A.: We thank the reviewer for the positive comments and assessment. We have thoroughly revised and edited the abstract passages to clarify them (see tracked changes).